# STRUCTURAL CAUSAL INTERPRETATION THEOREM

## ABSTRACT

Human mental processes allow for qualitative reasoning about causality in terms of mechanistic relations of the variables of interest, which we argue are naturally described by structural causal model (SCM). Since interpretations are being derived from mental models, the same applies for SCM. By defining a metric space on SCM, we provide a theoretical perspective on the comparison of mental models and thereby conclude that interpretations can be used for guiding a learning system towards true causality. To this effect, we present a theoretical analysis from first principles that results in a human-readable interpretation scheme consistent with the provided causality that we name structural causal interpretations (SCI). Going further, we prove that any existing neural induction method (NIM) is in fact interpretable. Our first experiment (E1) assesses the quality of such NIM-based SCI. In (E2) we observe evidence for our conjecture on improved sample-efficiency for SCI-based learning. After conducting a small user study, in (E3) we observe superiority in human-based over NIM-based SCI in support of our initial hypothesis.

## 1 INTRODUCTION

There has been an exponential rise in the use of machine learning, especially deep learning in several real-world applications such as medical image analysis (Ker et al., 2017), particle physics (Bourilkov, 2019), drug discovery (Chen et al., 2018) and cybersecurity (Xin et al., 2018) to name a few. While there have been several arguments that claim deep models are interpretable, the practical reality is much to the contrary. The very reason for the extraordinary discriminatory power of deep models (namely, their depth) is also the reason for their lack of interpretability. To alleviate this shortcoming, interpretable machine learning (Chen et al., 2019; Molnar, 2020) has gained traction to explain algorithm predictions and thereby increase the trust in these learned models.

In their seminal book, Pearl & Mackenzie (2018) argue that causal reasoning is the most important factor for machines to achieve true human-level intelligence. The same has been pointed out recently by Hofman et al. (2021) who argue that systems that are efficient in both causality and interpretations are need of the hour. Questions of the form "What if?" and "Why?" have been shown to be used by children to learn and explore their external environment (Gopnik, 2012; Buchsbaum et al., 2012) and are essential for human survival (Byrne, 2016). This makes understanding and reasoning about causality an inherently important problem.

While acknowledging the difficulty of the problem, we push in this direction pragmatically by presenting the first work on causal interpretations—interpreting a (deep) causal induction method by grounding it in its causal semantics, which we call *structural causal interpretations* (SCI). We show how neural causal induction models due to their causal semantics are interpretable, but then go another step, by also suggesting how these interpretations, if available, can be used in the first place to improve on our models understanding, thereby also establishing their importance in the process. The motivation behind this work is to move beyond black-box heat map based methods in explainable artificial intelligence. Although there exist some works such as explanatory interactive learning (XIL) (Teso & Kersting, 2019) and Clever-Hans methods (Lapuschkin et al., 2019; Stammer et al., 2021) that do move beyond such heat maps, we can go a step further and provide structured causal interpretations. XIL methods can fix Clever-Hans like moments but are overly dependent on the expert provided explanations. SCIs provide an elegant way to circumvent the Clever-Hans moments while also avoiding external factors such as expert explanations. Thus, we set the foundation for further research into making neural models, without initial causal semantics, possibly more causal.

Overall, we make the following contributions: (1) Starting from first principles involving mental models, causality and interpretations, we establish SCI theoretically to provide human-understandable interpretations from causal models, (2) we prove that SCI are readily available and that in fact any neural induction model (NIM) is SCI-interpretable, (3) we show empirically for three example members of NIM (Zheng et al. (2018),Goudet et al. (2018),Yu et al. (2019)) multiple example SCIs and assess their quality relative to the true underlying causality, (4) we show that existing SCIs can help in improving the learning-based causal induction, (5) we conduct a small human study to judge the qualitative causal structure for some graphs and show that human-based SCIs are close to the ground truth interpretations. We make our code publicly available.[1]

## 2 BACKGROUND AND RELATED WORK

Let us briefly review the background on a recent stride in learning that proposes a continuous formulation of data-driven estimation of (causal) graph structures, then the formalism of structural causal models and how estimation of respective dependency terms is conducted, and finally consider established works on the abstract high level ideas of explanations and interpretations.

**Learning Directed Acyclic Graphs.** Induction of inter-variable relationships based on available data lies at the core of most scientific endeavour (Penn & Povinelli, 2007). The sub-class of relation structures known as Directed Acyclic Graphs (DAG) is being focussed here due to its representational role in causality (Pearl, 2009; Peters et al., 2017). Unfortunately, due to the combinatoric nature of the problem setting, learning DAGs from data is recognized to be an NP-hard problem (Chickering et al., 2004). In their seminal work, Zheng et al. (2018) were able to re-formulate the traditional view into a continuous shape such that any non-convex optimization module can be applied to tackle the graph estimation problem. The authors propose the general formulation,

$$\min_{\boldsymbol{W} \in \mathbb{R}^{d \times d}} \quad f(\boldsymbol{W}) \quad \text{subject to} \quad h(\boldsymbol{W}) = 0, \tag{1}$$

where $f$ is a data-based score, e.g. in Zheng et al. (2018) a regularized least-squares loss is applied assuming a sparse linear SCM i.e., $f(\boldsymbol{W}) = ||\boldsymbol{X} - \boldsymbol{X}\boldsymbol{W}||_F^2 + ||\boldsymbol{W}||_1$, and $h$ is a smooth function with a kernel (or null space) that only contains acyclic graphs, $h(\boldsymbol{W}) = 0 \iff \boldsymbol{W}$ is acyclic. For the acyclicity constraint, different variations of the same continuous counting mechanism have been proposed, e.g., Zheng et al. (2020) proposed $h(\boldsymbol{W}) = \text{tr}(e^{\boldsymbol{W} \circ \boldsymbol{W}}) - d$ while Yu et al. (2019) proposed $h(\boldsymbol{W}) = \text{tr}[(\boldsymbol{I} + \boldsymbol{W} \circ \boldsymbol{W})^m] - m$, unfortunately, both suffer from cubic runtime-scalability in the number of graph nodes $O(d^3)$. While the aforementioned works have focussed on data originating from (non-linear transformation) of linear SCM, there exists yet another sub-class of DAG-learning methodologies that focuses on more general causal inference. Ke et al. (2019) made use of data from the first two levels of Pearl's Causal Hierarchy (PCH) (Pearl, 2009; Bareinboim et al., 2020), namely observational and interventional, to update their graph estimate $\boldsymbol{W} = G(\mathfrak{C})$ while using masked neural networks to mimic the structural equations $f_i = \text{MLP}_{\boldsymbol{\theta}}(\text{pa}(X_i))$ in order to maximize the likelihood of the data under given parameterization. Brouillard et al. (2020) follows the same idea of leveraging causal information, e.g. interventional data, for overcoming identifiability issues while staying close to the continuous optimization formalism introduced in 1.

**Causal Models and Effect Estimation.** Following Peters et al. (2017), a Structural Causal Model (SCM) is defined as $\mathfrak{C} := (\mathbf{S}, P_{\mathbf{N}})$ where $P_{\mathbf{N}}$ is a product distribution over noise variables and $\mathbf{S}$ is defined to be a set of $d$ structural equations

$$X_i := f_i(\text{pa}(X_i), N_i), \quad \text{where } i = 1, \dots, d \tag{2}$$

with $\text{pa}(X_i)$ representing the parents of $X_i$ in graph $G(\mathfrak{C})$. An SCM $\mathfrak{C}$ induces a DAG $G$ with edges $(i, j) \in V \times V$ meaning $i$ causes $j$, induces an observational/associational distribution $p^{\mathfrak{C}}$, it can be intervened upon using the *do*-operator and thus generate interventional distributions $p^{\mathfrak{C};do(\dots)}$ and furthermore given some observations $\mathbf{v}$ can also be queried for interventions within a system with fixed noise terms amounting to counterfactual distributions $p^{\mathfrak{C}|\mathbf{V}=\mathbf{v};do(\dots)}$, that is, $\mathfrak{C} \implies$ PCH. To query for samples of a given SCM, the structural equations are being simulated sequentially following the underlying causal structure starting from the independent, exogenous variables. The dependency terms provided by the structural equations $f_i$ defines the translation between domains

---

[1] https://anonymous.4open.science/r/Structural-Causal-Interpretation-Theorem-D5C0

of the cause and the effect i.e., the causal effect $\mathfrak{g}(X \to Y)$. In most settings, this causal effect, e.g., of a medical treatment onto the patient's recovery, is the sought quantity of interest. If interventions are admissible, then the average causal (or treatment) effect (ACE/ATE) within a binary system is defined as a difference in interventional distributions: $\mathfrak{g}(X \to Y) := \text{ACE}(X, Y) = \mathbb{E}[Y \mid do(X = 1)] - \mathbb{E}[Y \mid do(X = 0)]$. The ACE overcomes the fundamental problem of causal inference, which states the restriction that individual-level causal effects cannot be estimated as we cannot observe counterfactuals, but still suffers from the problem of identifiability i.e., that we usually don't even have access to interventions and thus need to resort to other causal knowledge (e.g. structural knowledge on confounders).

**Explanations and Interpretations in Deep Learning.** A great body of work within deep learning has provided visual means for explanations of how a neural model came up with its decision i.e., importance estimates for a model's prediction are being mapped back to the original input space e.g. raw pixels (Selvaraju et al., 2017; Sundararajan et al.; Schwab & Karlen, 2019; Schulz et al., 2020). Stammer et al. (2021) argue that such explanations are insufficient for any task that requires symbolic-level knowledge while comparing the existing state of explanations to children that are only able to point fingers but lack articulation. While agreeing with the aspect that simple heat-map explanations are indeed insufficient for reasoning tasks, we consider the higher order task of understanding the meaning of a matter (why) to be akin to what is meant by the word interpretation, while an explanation (invariant to its instantiation) refers to identifying the matter of interest (what). Seen from this perspective, explanations are a crucial sub-process within interpretation. Thereby what the authors in (Stammer et al., 2021) seek is in fact an interpretation i.e., the articulation by the child which also takes into account the causal dynamics of the scene.

## 3 STRUCTURAL CAUSAL INTERPRETATIONS

We first provide a big picture view on the way human mental models can be expressed in terms of SCMs and how only the true underlying causality is of interest during optimization. Then we provide intuition with our leading example on how causal semantics can be queried for answers that interpret their response based on the underlying causality before formalizing our main results.

### 3.1 HUMAN MENTAL MODELS AND THE IMPORTANCE OF TRUE CAUSALITY

To illustrate one's thought process about the understanding of the world dynamics is argued to lie at the core of a human mental model (Simon, 1961; Nersessian, 1992; Chakraborti et al., 2017). Although the concept of a mental model might contain circular and abstract terms like explanation and interpretations of its own, assuming the world dynamics to be governed by causality[2] we observe that humans are capable of modelling both causal relationships between system variables and additionally information on the strength of said relationship. Consider the following real world example, at any given time a human has a state of overall health (relating to fat-muscle ratio, allergies and diseases, etc.) and mobility (relating to the general freedom and flexibility of movement, e.g., a gymnast is more mobile than the average person). Now, a human can argue the following (1) that mobility is being (partially) directly caused by something, in this case for instance health, e.g., overweight will decrease the range of motion of any individual (2) that different events (even if mediated via the causal variable health) can have a different severity of impact e.g., that an average car accident causes more harm to the mobility of an individual than does an average workout session at the gym does good. A natural candidate for capturing such properties formally are Structural Causal Model (SCM), thereby we hypothesize the following:

**Hypothesis 1 (Causal Mental Model Conversion (CMMC)).** *The parts of the human mental model that are being used for encoding the causal relationships of reality's variables can be formally captured by a corresponding Structural Causal Model (see Sec.2 [2]).*

The CMMC hypothesis suggests that we can make use of SCM for expressing portions of human understanding and intuition within a mathematical language. Given such a formulation with SCM, the implications allow for some interesting observations to be made with the key observation being that SCM live in a metric space:

---

[2]For an extended treatise on why this is a sensible assumption consider (Pearl, 2009; Peters et al., 2017).

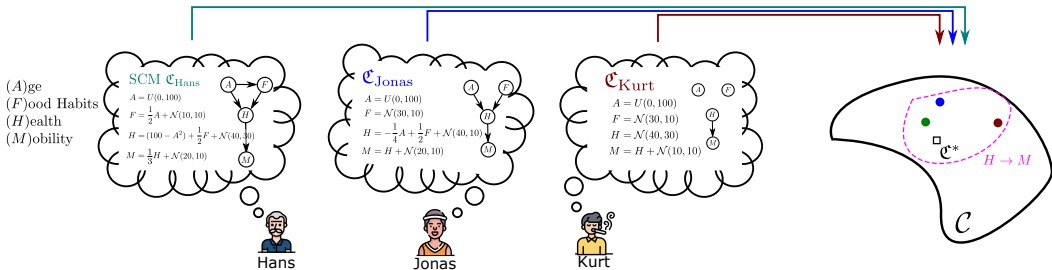

**Figure 1: Structural Causal Model Perspective onto Human Mental Models.** The CMMC hypothesis (see Hyp.1) states that human mental model, which are capable of both modelling causal relations (overall health affects mobility, $H \rightarrow M$) and the strength of such a relation (an average car accident has more negative impact, $\mathfrak{g}(H \rightarrow M)$, than an average workout has positive), can be naturally formalized within corresponding SCM. Furthermore, SCM live on a metric space $(\mathcal{C}, d)$ (see Thm.1) and can overlap (e.g. in pink, all humans agree on $H \rightarrow M$), therefore, most causal estimates will in fact deviate from the underlying truth, $\hat{\mathfrak{C}} \neq \mathfrak{C}^*$. (Best viewed in color.)

**Theorem 1** ($n$-**SCM Metric Space**)**.** *Let* **S** *be a set of structural equations* $X_j := f_j(\mathrm{pa}_j, N_j)$ *and* $P_{\mathbf{N}}$ *a product distribution, then* $\mathcal{C}_n = \{(\mathbf{S}, P_{\mathbf{N}}) : |\mathbf{S}|=n\}$ *defines the set of all constructable* $n$-*SCMs. Furthermore, define* $d : (\mathfrak{C}_n^1, \mathfrak{C}_n^2) \mapsto \sum_{i \neq j} |\mathfrak{g}(\mathbf{S}_i^1(j)) - \mathfrak{g}(\mathbf{S}_i^2(j))| + q(P_{\mathbf{N}}^1, P_{\mathbf{N}}^2)$ *where* $q := \sqrt{JSD}$ *is the Jensen-Shannon-Metric on the product distributions and* $\mathfrak{g}(j \rightarrow i) \in \mathbb{R}$ *defines the (expected) causal effect of* $j$ *on* $i$ *where* $\mathbf{S}_i(j)$ *refers to the isolated dependency term within the struct.eq. i.e.,* $f_i(\mathrm{pa}_i, N_i) = f_N(N_i) + \sum_{j \in \mathrm{pa}_i} f_{i,j}(j)$. *Then* $(\mathcal{C}_n, d)$ *defines a metric space.*

Because of space restrictions we provide the proof to Thm.1 and all subsequent mathematical results within the supplementary section. This metric space describes the set of all SCM, which themselves are a set of functions in addition to a product distribution, and they can all be compared against each other by the nature of their formulation. Under the CMMC hypothesis, it follows from Thm.1 that in fact parts of the original *human mental models* can be compared against each other thereby capable of dis-/agreeing. Interestingly, this comparability of SCM/human mental models allows for a trivial but crucial insight: if we additionally make the sensible assumption that there exists an 'objective' true SCM within the metric space, $\mathfrak{C}^* \in \mathcal{C}$, then most hypothesizable SCM will in fact be wrong. *This insight tells us that causality itself is not what will help in improving our models but being close to the true causality will.* Note that while we considered an 'objective true causality', human mental models are of subjective nature. We make the argument that having access to many SCM-encodings of subjective human mental models can ultimately lead in their overlap-agreement to (parts of) the objective true causality i.e., a higher quantity equates to a higher likelihood. All the established key concepts thus far are being visually illustrated within Fig.1. For machine learning research, observing that only a convergence towards true causality is beneficial might not be surprising, but observing that a set of 'subjective' SCM can provide information is indeed surprising if we consider interpretations. Interpretations are derivable from mental models and thus implicitly contain partial information on the latter's world dynamics representation (Chakraborti et al., 2017). Subsequently, if we can query humans for their interpretations, then we can make use of the acquired data, which by definition contain causal information, to improve existing ML models by providing them with (parts of) the true causality. In the spirit of the motivation provided by Pearl that all ML models should be aware of cause and effect. The benefits of an approach using interpretations derived from SCM are two-fold (1) that by construction they are human understandable allowing for interpretable ML in which models can reason about the learnt and (2) that the models themselves become better which is beneficial to any downstream-task. After establishing the connection between the formalism of causality and human mental models and using it for arguing that derived interpretations can be used to improve ML models, we now move onto the question of how to derive such interpretations from the causal semantics in the first place while alongside providing intuitive examples.

### 3.2 Answering Human Understandable Questions About Data Using Causality

Upon establishing that interpretations are derivable from SCM and should be used for improving ML models, this section provides an intuitive understanding of how data, queries and causal seman-

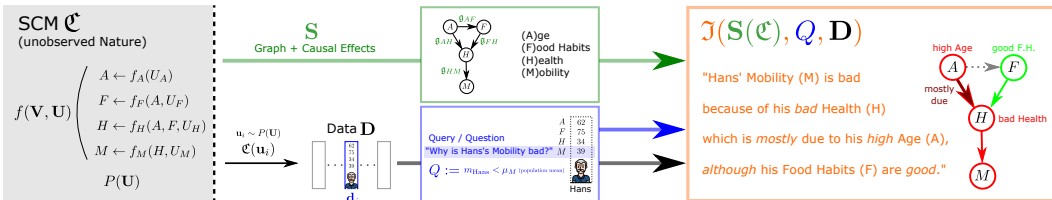

Figure 2: **Why Questions and Structural Causal Interpretations** The unobserved nature depicted by SCM $\mathfrak{C}$ on the left. The why-question $Q$ is an individual-level query derived from some population $\boldsymbol{D}$ originating in $\mathfrak{C}$. The graphical structure of $\mathfrak{C}$ alongside its causal effect terms induces the sub-structure $\boldsymbol{S}(\mathfrak{C})$. Following Thm.2, we arrive at a human expressible answer: the Structural Causal Interpretation $\mathfrak{I}(\boldsymbol{S}(\mathfrak{C}), Q, \boldsymbol{D})$. Our paper's lead example concerning Hans's mobility is being highlighted alongside the intuitive-level computation in the graph (right). (Best viewed in color.)

tics are being used in conjunction for generating interpretations both for explanatory and learning purposes. Figure 2 is being used as lead illustration for the interpretability of causal models alongside different real-world inspired examples. In the following, we consider (single) why-questions as goal to model inference as they induce answers that reason e.g. "...*because* ..." which are generally accepted as inferences about causal quantities (Pearl, 2009). To follow suit with the previously established examples around personal health, we could pose a question like *"Why is Hans's Mobility bad?"* after observing the state of causal variables for the individual named Hans, with the causal attributes being age, nutrition/food habits, overall health and mobility respectively. Given a SCM that represents age and nutrition as being causal for the health and subsequently mobility, while nutrition is generally also being affected by age, one can conclude the following human understandable interpretation as an answer to the initial question about Hans's rather immobile state:

**Interpretation 1** (Hans's Questioned State of Mobility). *"Hans's Mobility is bad because of his bad Health which is mostly due to his high Age although his Food Habits are good."*

This causal interpretation of Hans's personal state captures both reasons and importances, as a consequence of the CMMC hypothesis (see Hyp.1) established previously. More importantly, Int.1 has been constructed automatically from the available data and causal knowledge in response to the query. As a first observation towards the interpretation construction, note that the why-query contains a relative notion *"why ... **bad**?"* that implicitly compares the individual Hans to the questionee's available data set i.e., remaining population. As a second key idea, note the trivial conclusion that by definition there can only exist a causal effect from some variable to another if and only if there is a causal directed path between the variables. As a third and final concept, consider the difference in causal effect i.e., that some variables exert more influence than others. Unifying these three building principles in a formal manner, to be discussed in the subsequent section, allows for the generation of interpretations like Int.1. That is, our SCM tells us that mobility of any individual will decrease with decreasing overall health and upon observing that Hans's health has been "bad" in the first place, we can conclude that the questioned state of mobility is due to Hans's health. We can now increase the granularity and reason recursively that, yet again, Hans's health is such due to his age and nutrition but this time we can introduce two more detailed distinctions. One, that Hans's nutrition is actually above average and therefore not the reason for his immobility. Second, that age's causal effect wages in more than that of nutrition, as a general statement of the given SCM, thus the asymmetry in attribution ("mostly due"). Conclusively, a **Structural Causal Interpretation (SCI)** depends on a measure of relativeness on a given data population, as implied by the given why-question, and on the structural equations provided by the assumed SCM[3]. It is worthwhile noting that the natural language choice of words to express the interpretation is not implied by the form of the SCI e.g., while Hans's mobility is said to be "bad", a car's mileage is considered to be "low" rather than "bad". On another note, similar to how Pearl has argued that Bayesian Networks (BN) are a suitable representation scheme for causality (Pearl, 2009; 2011) because of BNs being direct representations of the world and not of reasoning processes, this work poses a parallel to the 'representation scheme' of interpretability using causal semantics. Interpretability becomes a part of the causal nature of the modelled. In the following, we mathematically formalize the established intuition on SCI and then proceed with the theoretical analysis of SCI-implications.

---

[3]The naming resembles that of SCM because of the importance of the structural causal information for generating any specific interpretation.

### 3.3 Mathematical Foundations of Structural Causal Interpretations

Interpretations as established in the previous section (see Int.1) act in response to a query. The queries or questions we pose are single-attribute assertions relative to a population that correspond to the causal inquiry "why?", formally:

**Definition 1** (**Single-Why Question**). *Let $x_i \in D(X)$ be a scalar instance of $X$, let $\mu_X$ be some population score (e.g. the mean of a sample $\mu_X := \frac{1}{n}\sum_j^n x_j$) and let $R$ be a binary ordering relation (e.g. $R \in \{<,>\}$), then a single-why question concerning $X$ is a true assertion $Q_X = R(x_i, \mu_X)$.*

Using Def.1, the query concerning our lead example from the previous section used for Int.1 is expressed as $Q_{\text{Hans}} := m_{\text{Hans}} < \mu_M$ where $\mu_M$ is the average mobility of an individual. The unification of the previously established three guiding ideas for generating interpretations (causal connection, strength and relativeness to population) can be formalized in a set of logic rules:

**Proposition 1** (**First-Order Logic Rules for Interpretations**). *Let $\mathfrak{g}(X{\to}Y) \in \mathbb{R}$ be a causal effect estimator, let $s(x) \in \{-1,1\}$ be the sign of a scalar, let $\mathcal{Z}_X = \{|\mathfrak{g}(Z{\to}X)| : Z \in \text{pa}_X\}$ be the set of absolute parental causal effects onto $X$, and $\mu_X, R$ as in Def.1. Furthermore, a rule indicator is defined as $\mathbf{1}_j(X){=}(-)1$ signalling with which relation (if) rule $j$ applies to input $X$ and $0$ otherwise. Then for any pair $X \in V, Y \in \text{pa}(X)$ and some individual-scalar $y \in D(Y)$,*

*(R1) Excitation:*    $R_1 {\neq} R_2 \implies [R_1(s(\mathfrak{g}(Y{\to}X)),0) \wedge [R_2(y,\mu_Y) \vee R_1(x,\mu_X)]]$,

*(R2) Inhibition:*    $R_1 {\neq} R_2 \implies [R_1(s(\mathfrak{g}(Y{\to}X)),0) \wedge R_1(y,\mu_Y) \wedge R_2(x,\mu_X)]$,

*(R3) Preference:*    $|\mathcal{Z}_X| > 1 \implies [Y \iff \arg\max_{Z \in \mathcal{Z}_X} Z]$,

*define a rule-set function $\mathfrak{R}(Y{\to}X){\in}\{-1,0,1\}^3$ indicating for each rule $j$ if and how the causal relation $Y{\to}X$ satisfies that rule (e.g. for R1 either over- or under-excitation). For any causal scenario $C_{XY}{:=}(\mathfrak{g}(Y{\to}X), y, x, \mu_X, \mu_X)$ excitation (R1) and inhibition (R2) are mutually exclusive.*

The three rules in Prop.1 are naturally derived from the principles established in the previous section. The rule-set function $\mathfrak{R}$ forms the core of the interpretation process which we finally formalize after introducing one last mathematical concept to provide for the causal semantics of the interpretation:

**Definition 2** (**Causal Effect Matrix (CEM)**). *Let $\mathfrak{C}$ be a SCM and $\mathfrak{g} : |V| \times |V| \to \mathbb{R}$ the causal effect function, we refer to $\boldsymbol{S}(\mathfrak{C}) := (\mathfrak{g}(i \to j))_{ij} \in \mathbb{R}^{|V| \times |V|}$ as the causal effect matrix of SCM $\mathfrak{C}$. As a special case, if $\boldsymbol{S}(\mathfrak{C}) \in [0,1]^{|V| \times |V|}$, then we call the resulting matrix the sign of the SCM.*

It is important to note that the CEM in Def.2 is a sub-structure to the SCM i.e., it contains less information. The CEM eventually enables the unification of existing approaches from DAG-learning into a family of interpretable models. These very interpretations that pose one of our theoretical main results, Structural Causal Interpretations, we finally formalize and prove as a recursion:

**Theorem 2** (**Structural Causal Interpretations**). *Let $Q_X, \boldsymbol{S}(\mathfrak{C})$, and $\boldsymbol{D} \in \mathbb{R}^{n \times d}$ be a single-why question on variable $X$, a CEM, and a data matrix respectively. Furthermore, let $\oplus_{i=1}^n v_i = (v_1, \dots, v_n)$ denote concatenation and $\mathfrak{R}$ be the rule-set function from Prop.1. Then a Structural Causal Interpretation is being defined recursively as*

$$\mathfrak{I}(Q_X, \boldsymbol{S}(\mathfrak{C}), \boldsymbol{D}) = (\bigoplus_{Z \in \text{pa}(X)} \mathfrak{R}(Z \to X), \bigoplus_{Z \in \text{pa}(X)} \mathfrak{I}(Q_Z, \boldsymbol{S}(\mathfrak{C}), \boldsymbol{D})) \tag{3}$$

*with the base case being evaluated at the roots of the causal path to $X$, that is,*

$$\exists Z \in V : (Z \to \cdots \to X \wedge \text{pa}(Z) = \emptyset \iff \mathfrak{I}(Q_Z, \boldsymbol{S}(\mathfrak{C}), \boldsymbol{D}) = \emptyset). \tag{4}$$

*The recursive algorithm for computing $\mathfrak{I}(Q_X, \boldsymbol{S}(\mathfrak{C}), \boldsymbol{D})$ in Eq.3 terminates on any input.*

Intuitively, the construction of SCI in Thm.2 considers maximum depth interpretations i.e., the recursion is being rolled out along the possible causal paths that converge into the query-variable until the exogenous root variables are being reached. Thereby, SCI implicitly define a notion of detail to the interpretation based on the rollout-depth of the recursion. Considering again our lead example in Int.1 from the previous section, the maximum-depth interpretation would also contain an argument on Hans's Age and Food Habits relation e.g. '[...] good Food Habits because of his high

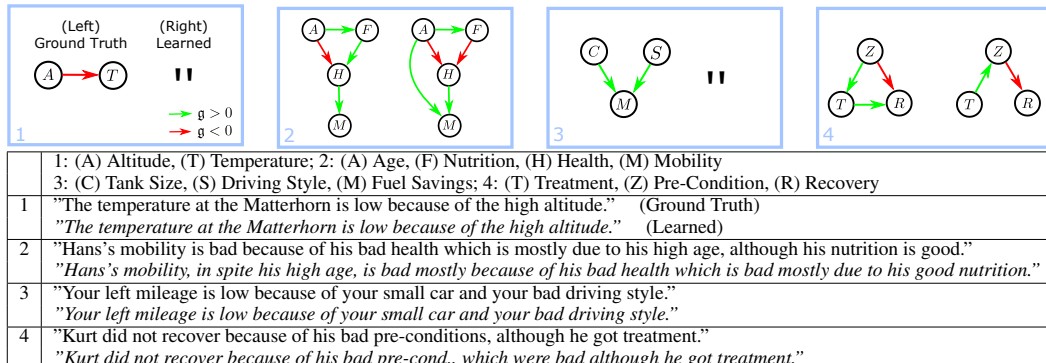

| | 1: (A) Altitude, (T) Temperature; 2: (A) Age, (F) Nutrition, (H) Health, (M) Mobility |
|---|---|
| | 3: (C) Tank Size, (S) Driving Style, (M) Fuel Savings; 4: (T) Treatment, (Z) Pre-Condition, (R) Recovery |
| 1 | "The temperature at the Matterhorn is low because of the high altitude."   (Ground Truth) |
| | *"The temperature at the Matterhorn is low because of the high altitude."*   (Learned) |
| 2 | "Hans's mobility is bad because of his bad health which is mostly due to his high age, although his nutrition is good." |
| | *"Hans's mobility, in spite his high age, is bad mostly because of his bad health which is bad mostly due to his good nutrition."* |
| 3 | "Your left mileage is low because of your small car and your bad driving style." |
| | *"Your left mileage is low because of your small car and your bad driving style."* |
| 4 | "Kurt did not recover because of his bad pre-conditions, although he got treatment." |
| | *"Kurt did not recover because of his bad pre-cond., which were bad although he got treatment."* |

Table 1: **Quality of Learned Interpretations.** Ground Truth SCM (left;normal) versus NOTEARS from Zheng et al. (2018) (right;italics). The considered Why-questions can be found pronounced as well as formal within Fig.2 respectively. Subtle differences between interpretations exist e.g., the last interpretation is right on the top-level but for the wrong reasons ($T \to Z$ instead of $T \to R$).

Age (life-time experience)'. At each recursion step, the satisfaction of the rule-set function is being examined which ultimately dictates the reading of the interpretation. In a nutshell, the why-question determines the starting point for the SCI recursion while the given causal semantics and data provide for the interpretation specific to the model and the available information. In the following, we make the leap towards connecting interpretability with existing methodologies from the DAG learning literature by first introducing formally the family of data-driven CEM estimators:

**Definition 3 (Family of CEM Estimators).** *Let $L : \mathcal{D}(\mathfrak{C}) \times \mathcal{H} \to \mathbb{R}$ be a scalar function on the data $\boldsymbol{D} \in \mathcal{D}(\mathfrak{C})$ generated from some underlying SCM $\mathfrak{C} = (\boldsymbol{S}, P_{\boldsymbol{N}})$ and on some DAG-structured hypothesis space $\mathcal{H}$ whose elements are defined as*

$$
\begin{matrix}
\to & X_1 & \dots & X_j & \dots & X_n \\
X_1 \\
\vdots
\end{matrix}
\begin{pmatrix}
0 & \dots & f_{1j} & \dots & f_{1n} \\
\vdots & & \vdots & & \vdots
\end{pmatrix}
\iff
\tag{5}
$$

*with $n = |\boldsymbol{S}|$ and functions $f_{ij}$ describing $i \to j$ with a scalar $f(x_i) \in \mathbb{R}$. The family of Causal Effect Matrix (CEM) estimators encompasses any learner that solves $H^* = \arg\min_{H \in \mathcal{H}} L(\boldsymbol{D}, H)$.*

Def.3 provides a big picture view on methods that perform causal induction and more importantly it argues that *any induction performed on the restricted hypothesis space of DAGs is in fact a causal induction* because any resulting hypothesis respects the structure of a SCM and will thus, by construction, resemble some SCM $\mathfrak{C} \in \mathcal{C}$ even if this SCM does not resemble the sought SCM $\mathfrak{C}$ which generated the available data $\boldsymbol{D} \in \mathcal{D}(\mathfrak{C}^*)$ i.e., $\mathfrak{C}^* \neq \mathfrak{C}$. This big picture view is inline with (Pearl & Mackenzie, 2018) where a next generation of learning systems capable of reasoning about cause-effect relations opposed to simple correlations is being endorsed. With Def.3 we make two key observations, firstly, that methods that estimate an SCM are more powerful than CEM-estimator:

**Proposition 2 (SCM- and CEM-Estimators).** *Any SCM estimator is a CEM estimator.*

Thereby, additional structural/causal information will only benefit the modelling procedure. The second consequence of Def.3 poses our third and final main result in that a host of available (neural) models from the DAG-learning literature are all interpretable:

**Theorem 3 (Neural Induction Models are Interpretable).** *Algorithms for (causal) DAG-induction from data (e.g. NT (Zheng et al., 2018), CGNN (Goudet et al., 2018), DAG-GNN (Yu et al., 2019), NCM (Ke et al., 2019)) can generate Structural Causal Interpretations.*

Thm.3 is important as it tells us that the causal reasoning, on at least the level of CEM, performed by any model-type (includes neural network based models) can be queried for causal interpretations in the sense of SCI (Thm.2). While the ability to provide answers to questions in an expressive manner which captures what has been learned by the model is of high intrinsic value, in the following we consider how existing interpretations can be exploited for use as a means to improve model

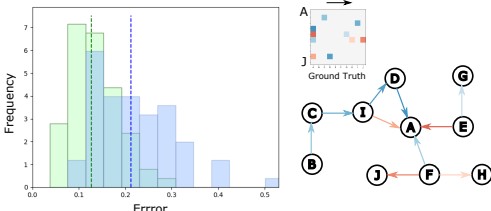 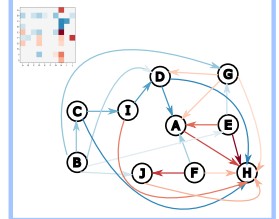 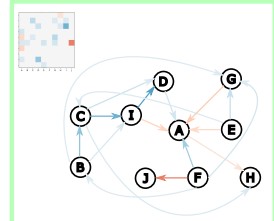

Figure 3: **Interpretations Improve Graph Induction.** On the left a comparison of the error distributions when performing graph induction with and without existing interpretations. Alg.1 with $\mathcal{M}_{\boldsymbol{\theta}} = $ NOTEARS is being applied. The number of falsely inferred graph links is being being reduced significantly as predicted by Conj.1. Technical details in Appendix. (Best viewed in color.)

learning in the first place. The following approach can be seen as an 'interpret to learn'-approach as the interpretations provided by the model are being evaluated against existing interpretations (possibly provided by humans) for improving the overall estimate of the underlying causality. It can also be seen as a regularizer on the model since it can extend on any loss formulation as long as the interpretation process $\mathcal{I}$ (usually $\mathcal{I} = \mathfrak{I}$ where $\mathfrak{I}$ refers to Eq.3) can be incorporated into the corresponding optimization routine (i.e., differentiability in the case of neural approximations).

**Conjecture 1** (**SCI Regularization**). *Let $L$ be defined as in Def.3 and let $H^*$ be the true CEM. Regularizing $L_r := L + \frac{1}{n} \sum_i ||\mathcal{I}(Q_X, H, \boldsymbol{D}) - \boldsymbol{I}_i)||_2^2$ with existing and overlapping SCI $\boldsymbol{I}$ leads to a more optimal solution $|\arg\min_{H \in \mathcal{H}} L_r(\boldsymbol{D}, H) - H^*| < |\arg\min_{H \in \mathcal{H}} L(\boldsymbol{D}, H) - H^*|$.*

The interpretations contain information about their respective CEM. While CEM are hard to estimate and usually an integral part of the sought quantity of interest, corresponding (approximate) interpretations are often times readily available and can thus be used. The regularization term introduced in Conj.1 thereby penalizes CEM that would not be able to account for certain explanations to the posed single-why questions. Furthermore, Conj.1 poses an interesting direction of future research concerning more general classes of causal models that could enable a tighter in-

---

**Algorithm 1** Dual Learning: Induct & Interpret

**Input**: Data $\boldsymbol{D}$, Inductor $\mathcal{M}_{\boldsymbol{\theta}}$, Interpret. $\mathbf{I}$, Optimizer $\mathcal{O}$
**Output**: CEM $H$, Causal Graph $G$

1: Let $H \leftarrow \boldsymbol{0}$
2: **while** $i \leq |\mathbf{I}|$ **do**
3:     $H, l \leftarrow \mathcal{M}_{\boldsymbol{\theta}}(\mathbf{D})$ {induction from data}
4:     $Q_X, \boldsymbol{I}^* \leftarrow \boldsymbol{I}_i$
5:     $\boldsymbol{I} \leftarrow \mathcal{I}(Q_X, H, \boldsymbol{D})$ {estimate's interpretation}
6:     $l_{\boldsymbol{I}} \leftarrow ||\hat{\boldsymbol{I}} - \boldsymbol{I}^*)||_2^2$ {compare}
7:     $\boldsymbol{\theta} \leftarrow \mathcal{O}(l, \boldsymbol{\theta})$ {parameter update}
8: **end while**
9: $H \leftarrow \mathcal{M}_{\boldsymbol{\theta}}(\mathbf{D})$, and $G \leftarrow |\tanh(H)|$
10: **return** $H, G$

---

tegration between current practices in deep learning and causality, as in (Xia et al., 2021) for general Neural Causal Models (NCM) and (Zečević et al., 2021b) for Graph Neural Networks based NCM.

## 4 EMPIRICAL ILLUSTRATION

To assist our theoretical results we provide an empirical illustration. Due to space constraints, we choose NOTEARS (NT, Zheng et al. (2018)) as representative for NIM-based experiments and further only highlight relevant key results while pointing to the **extensive analysis in the Appendix**.

**Experiment 1: NIM-based SCI (Thm.3).** Tab.1 considers four data sets and single-why queries illustrated in Fig.2 i.e., Deutscher Wetterdienst (DW, Mooij et al. (2016)), Causal Health (CH, Zečević et al. (2021a)), Mileage (M), and Recovery (R, Charig et al. (1986)). The algorithmic interpretations are sensible but differ in terms of quality. The SCI (Thm.2) generated using the learned causal semantics match the given ground truth interpretations for the DW and M data sets, while differing only slightly for R and drastically for CH data sets. The violation of the linearity assumption might offer an NT-specific explanation. An extensive study is provided in the Appendix.

**Experiment 2: SCI-based Learning (Conj.1).** We use Alg.1 where we set $\mathcal{I} = \mathfrak{I}_{\boldsymbol{\psi}}$ with latter being a neural approximation to the SCI recursion from Eq.3 to assure differentiability. The graph

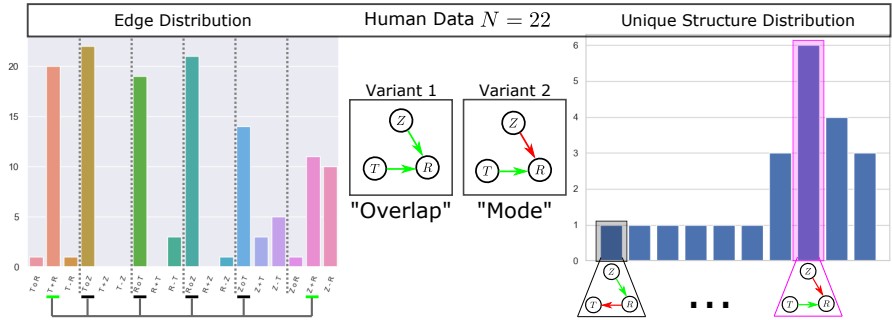

Figure 4: **Human CEM Agreement Variants.** We consider two variants of measuring agreement within the human causal induction data: overlap (H1) and mode (H2). The H1-CEM structure is a greedy aggregation that chooses the most-voted edge type for each edge. The H2-CEM is simply the most frequently occuring CEM structure. (Best viewed in color.)

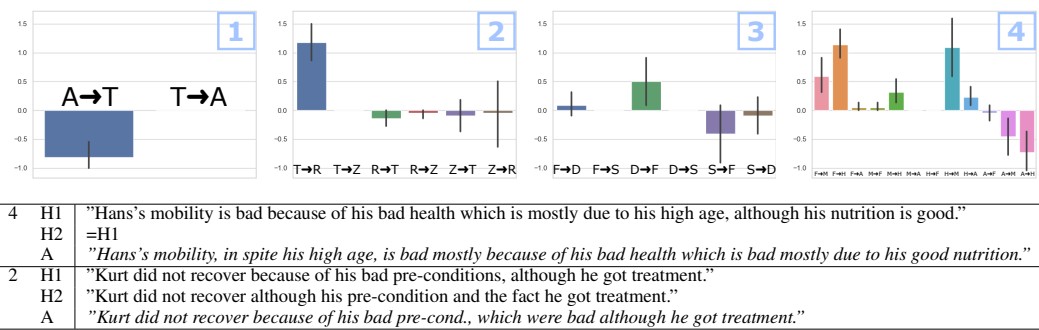

| 4 | H1 | "Hans's mobility is bad because of his bad health which is mostly due to his high age, although his nutrition is good." |
|---|----|---|
|   | H2 | =H1 |
|   | A  | *"Hans's mobility, in spite his high age, is bad mostly because of his bad health which is bad mostly due to his good nutrition."* |
| 2 | H1 | "Kurt did not recover because of his bad pre-conditions, although he got treatment." |
|   | H2 | "Kurt did not recover although his pre-condition and the fact he got treatment." |
|   | A  | *"Kurt did not recover because of his bad pre-cond., which were bad although he got treatment."* |

Table 2: **'Humans vs Algorithms'.** Top: Edge plots per example where the bars denote the average value of given relation and the errors confidence intervals. Bottom: The SCI generated for the two human variants (H1, H2 from Fig.4) against an algorithm representative (NT, Zheng et al. (2018)). Human interpretations are (near-)identical to the ground truth from Tab.1. (Best viewed in color.)

induction is being performed on Erdos–Renyi structures in a data-scarce setting with only 10 data samples per graph induction. Fig.3 shows our empirical results on the error distributions for all the graphs with the best improving optimization being highlighted. We observe an increase in sample-efficiency. The interpretations contain valuable information about the underlying CEM that allow reducing the search space, thereby favoring Conj.1. An extensive study is provided in the Appendix.

**Experiment 3: Algorithmic vs. Human Interpretations.** We let 22 human subjects judge the qualitative causal structure for each of the four examples (questionnaire in Appendix). In Fig.4 we consider two variants of measuring the structure that the humans agreed upon. In Tab.2 we show the quantitative edge plots (top) which expose common ground in the human-based inferences and the resulting SCI (bottom) which significantly outperform the algorithmic-based. The humans results are in support of the CMMC hypothesis (Hyp.1). An extensive study is provided in the Appendix.

## 5   CONCLUSIONS AND FUTURE WORK

Starting from first principles, we discussed the connection between human mental/thinking modes and SCMs (Hyp.1) which lead to the derivation of a metric space for SCMs (Thm.1). We then derived three basic rules (Prop.1) that jointly with why-questions (Def.1) allow for establishing *Structural Causal Interpretations* (SCI, Thm.2). We proved that any neural induction method (NIM) is thus interpretable (Thm.3). Empirically, we showed that the NIM-SCI are sensible (E1), that SCIs help in improving learning (E2; Conj.1), and finally a human case study that allows for comparing NIM- and human-based SCI (E3). Following, an extension of the query type (Def.1) and accordingly SCI-formalism (Thm.2) might allow for more complex interpretations aligned in spirit with the intuition behind the PCH-levels $\mathcal{L}_i$ (Pearl & Mackenzie, 2018; Bareinboim et al., 2020). Another natural followup is to perform a large-scale user study for curating a rich human-based SCI data set.

## ETHICS STATEMENT

With our work, we have shown that we can seamlessly encapsulate several (deep) causal induction methods inside a single interpretation framework that is consistent with the learned causal semantics. The major impact that our work pursues is to provide human understandable answers to human understandable questions while respecting the learned (or assumed) causality. This endeavour can have several implications on producing causally interpretable signatures, which could be predictive of the emergence and the success of new fields or discoveries.

Another concern can be the use of the human user study in our work. Although the questions asked are very general and do not pose any threat to the privacy of the participating users, we anonymize the responses before using them. To the best of our knowledge, our study does not raise any ethical, privacy or conflict of interest concerns and obliges common practices and standards for experimental setups in behavioral and social sciences.

## REPRODUCIBILITY STATEMENT

We provide the code and data sets to reproduce the reported results through an anonymous online repository. Lastly, details on the conducted user study along with the received and anonymized responses are also provided in the appendix.

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

# A    APPENDIX - STRUCTURAL CAUSAL INTERPRETATION THEOREM

We make use of this appendix following the main paper to provide the proofs to the main theorems and propositions in addition to further assisting theoretical results, alongside details on the regularization experiment and details on the execution of the human case study.

## A.1    PROOF FOR THEOREM 1

We argue strongly for the view that adequate or "true" causality is what learning systems should seek to model opposed to simply *any* causality derivable from the given data. In fact, we believe this misconception to be a strong oppressor of current research practices. An important consequence of the fact that different causal modellings exist, is that they become comparable in terms of their SCM specifications. The following theorem suggests that there exists always a well-defined metric space for SCMs of same size, allowing for a feasible distance computation between any two SCMs within.

**Theorem 1** ($n$-SCM Metric Space). *Let $\mathbf{S}$ be a set of structural equations $X_j := f_j(\mathrm{pa}_j, N_j)$ and $P_{\mathbf{N}}$ a product distribution, then $\mathcal{C}_n = \{(\mathbf{S}, P_{\mathbf{N}}) : |\mathbf{S}|=n\}$ defines the set of all constructable $n$-SCMs. Furthermore, define $d : (\mathfrak{C}_n^1, \mathfrak{C}_n^2) \mapsto \sum_{i \neq j} |\mathfrak{g}(\mathbf{S}_i^1(j)) - \mathfrak{g}(\mathbf{S}_i^2(j))| + q(P_{\mathbf{N}}^1, P_{\mathbf{N}}^2)$ where $q := \sqrt{JSD}$ is the Jensen-Shannon-Metric on the product distributions and $\mathfrak{g}(j \to i) \in \mathbb{R}$ defines the (expected) causal effect of $j$ on $i$ where $\mathbf{S}_i(j)$ refers to the isolated dependency term within the struct.eq. i.e., $f_i(\mathrm{pa}_i, N_i) = f_N(N_i) + \sum_{j \in \mathrm{pa}_i} f_{i,j}(j)$. Then $(\mathcal{C}_n, d)$ defines a metric space.*

*Proof.* The absolute difference on the real numbers is a metric (i.e., positive-definiteness, symmetry, and triangle-inequality hold) and $\mathfrak{g}(j \to i) \in \mathbb{R}$, furthermore, $q$ is chosen as the Jensen-Shannon-Metric by construction. Finally, a sum of two metrics is itself a metric. $\square$

## A.2    PROOF FOR PROPOSITION 1

The logic defined in Prop.1 posed the key to defining the actual interpretation scheme later on in Thm.2. It is sensibly defined to adhere to the causal semantics of the system and in the following we prove that the logic rules are also consistent w.r.t. to the causal semantics i.e., for any possible kind of causal scenario, expressed through $C_{XY}:=(\mathfrak{g}(Y \to X), y, x, \mu_X, \mu_X)$, we prove that either excitation (**R1**) or inhibition (**R2**) occur but never simultaneously, with preference (**R3**) optionally.

**Proposition 1** (First-Order Logic Rules for Interpretations). *Let $\mathfrak{g}(X \to Y) \in \mathbb{R}$ be a causal effect estimator, let $s(x) \in \{-1, 1\}$ be the sign of a scalar, let $\mathcal{Z}_X = \{|\mathfrak{g}(Z \to X)| : Z \in \mathrm{pa}_X\}$ be the set of absolute parental causal effects onto $X$, and $\mu_X, R$ as in Def.1. Furthermore, a rule indicator is defined as $\mathbf{1}_j(X)=(-)1$ signalling with which relation (if) rule $j$ applies to input $X$ and $0$ otherwise. Then for any pair $X \in V, Y \in \mathrm{pa}(X)$ and some individual-scalar $y \in D(Y)$,*

*(R1) Excitation:*    $R_1 \neq R_2 \implies [R_1(s(\mathfrak{g}(Y \to X)), 0) \land [R_2(y, \mu_Y) \lor R_1(x, \mu_X)]],$

*(R2) Inhibition:*    $R_1 \neq R_2 \implies [R_1(s(\mathfrak{g}(Y \to X)), 0) \land R_1(y, \mu_Y) \land R_2(x, \mu_X)],$

*(R3) Preference:*    $|\mathcal{Z}_X| > 1 \implies [Y \iff \arg\max_{Z \in \mathcal{Z}_X} Z],$

*define a rule-set function $\mathfrak{R}(Y \to X) \in \{-1, 0, 1\}^3$ indicating for each rule $j$ if and how the causal relation $Y \to X$ satisfies that rule (e.g. for **R1** either over- or under-excitation). For any causal scenario $C_{XY}:=(\mathfrak{g}(Y \to X), y, x, \mu_X, \mu_X)$ excitation (**R1**) and inhibition (**R2**) are mutually exclusive.*

*Proof.* We have to prove that any $C_{XY}:=(\mathfrak{g}(Y \to X), y, x, \mu_X, \mu_X)$ will trigger either **R1** or **R2** in any case but never simultaneously. For this, consider the encoding $\{< \mapsto 0, > \mapsto 1\}$ i.e., the relations are mapped to the binary numbers. Given some causal scenario $C_{XY}$, for both **R1** and **R2** we always have an evaluation of the triplet $T = (R(s(\mathfrak{g}(Y \to X)), 0), R(y, \mu_Y), R(x, \mu_X))$. It is easy to see that $|T| = 2^3 = 8$ is the total number of possible scenarios, where **R1** covers codewords $(010, 011, 100, 101, 000, 111)$ and **R2** covers the codewords $(001, 110)$, and together they cover all codewords $|\mathbf{R1}| + |\mathbf{R2}| = |T|$. Since any single scenario $C_{XY}$ is uniquely mapped to a codeword through its triplet, it will either trigger **R1** or **R2** but never fall into both in parallel. $\square$

### A.3 Proofs for Theorem 2 and Proposition 2

Thm.2, being our second and main contribution, suggests a construction scheme based on the three base logic rules of arguably axiomatic nature (Prop.1) for generating human-understandable answers to single why-questions (Def.1). The rules are being recursively applied to the given causal effect matrix estimation (Def.2), that is structure and parameterization, alongside data. The recursion for computing the SCI $\mathfrak{I}$ in Eq.6 terminates for any input tuple $(Q_X, \boldsymbol{S}(\mathfrak{C}), \boldsymbol{D})$.

**Theorem 2** (**Structural Causal Interpretations**). *Let $Q_X$, $\boldsymbol{S}(\mathfrak{C})$, and $\boldsymbol{D} \in \mathbb{R}^{n \times d}$ be a single-why question on variable $X$, a CEM, and a data matrix respectively. Furthermore, let $\oplus_{i=1}^{n} v_i = (v_1, \ldots, v_n)$ denote concatenation and $\mathfrak{R}$ be the rule-set function from Prop.1. Then a Structural Causal Interpretation is being defined recursively as*

$$\mathfrak{I}(Q_X, \boldsymbol{S}(\mathfrak{C}), \boldsymbol{D}) = (\bigoplus_{Z \in \mathrm{pa}(X)} \mathfrak{R}(Z \to X), \bigoplus_{Z \in \mathrm{pa}(X)} \mathfrak{I}(Q_Z, \boldsymbol{S}(\mathfrak{C}), \boldsymbol{D})) \tag{6}$$

*with the base case being evaluated at the roots of the causal path to $X$, that is,*

$$\exists Z \in V : (Z \to \cdots \to X \wedge \mathrm{pa}(Z) = \emptyset \iff \mathfrak{I}(Q_Z, \boldsymbol{S}(\mathfrak{C}), \boldsymbol{D}) = \emptyset). \tag{7}$$

*The recursive algorithm for computing $\mathfrak{I}(Q_X, \boldsymbol{S}(\mathfrak{C}), \boldsymbol{D})$ in Eq.3 terminates on any input.*

*Proof.* We have to show that the base case is being reached eventually for any input tuple $(Q_X, \boldsymbol{S}(\mathfrak{C}), \boldsymbol{D})$. The second tuple entry (right) in Eq.6 calls $\mathfrak{I}$ with a query on the parents of a given node $X$, that is, $Q_Z, Z \in \mathrm{pa}(X)$ and otherwise same arguments. Since any SCM $\mathfrak{C} = (\boldsymbol{S}, P(\boldsymbol{U}))$ is finite, i.e., the number of structural equations is an integer $|\boldsymbol{S}| = d \in \mathbb{N}$, and $\mathfrak{C}$ implies a DAG $G$, we are guaranteed to reach a root node $K$ for which $\mathrm{pa}(K) = \emptyset$ and since $K$ is an ancestor to $X$, it also holds that $K \to \cdots \to X$. This in fact is the requirement for reaching the base case. $\square$

Since the CEM is a sub-structure of the SCM as it "summarizes" the given structural equations to their causal effects for each of the edges within the SCM's induced graph, we can make the important observation that methods that in fact model the actual SCM (e.g. NCMs as in (Xia et al., 2021) or (Zečević et al., 2021b)) are more powerful than methods that model the sub-structure (CEM).

**Proposition 2** (**SCM- and CEM-Estimators**). *Any SCM estimator is a CEM estimator.*

*Proof.* Let $g : \mathcal{F}_j^i \mapsto \mathbb{R}$ represent the (expected) causal effect from $j$ to $i$ described by a functional form $f \in \mathcal{F}$. A structural equation can be split into its dependency terms, $f_i(\mathrm{pa}_i, N_i) = f_N(N_i) + \sum_{j \in \mathrm{pa}_i} f_{i,j}(j)$, such that the matrix $M = (\mathfrak{g}(f_{i,j}(j)))_{ij}$ is a CEM. $\square$

### A.4 Proof for Theorem 3

Thm.3 is a key theoretical result that suggests that any of the existing neural causal induction methods, independent of their assumptions[4] on the data-generating SCM, are in fact interpretable using the SCI from Thm.2. Thereby, Thm.3 provides a completely new perspective on both what (neural) causal induction methods are capable of simply due to the fact that they model causal relations and also more importantly how a human inspector can investigate the model's understanding of the given data in a humanly understandable manner, since SCI imply a direct translation to natural language (as is being detailed in below's section) they also imply transparent communication.

**Theorem 3** (**Neural Induction Models are Interpretable**). *Algorithms for (causal) DAG-induction from data (e.g. NT (Zheng et al., 2018), CGNN (Goudet et al., 2018), DAG-GNN (Yu et al., 2019), NCM (Ke et al., 2019)) can generate Structural Causal Interpretations.*

*Proof.* Causal semantics at the level of CEM or above (e.g. SCM, see Prop.2) allow by construction for SCI (Thm.2), what is left to show is that the mentioned neural induction methods at least estimate CEM. We start with our lead example method NT. We proceed with proving NT (Zheng et al., 2018) to be interpretable, the other proofs will be analogous. NT estimates a linear SCM whose structural

---

[4]An example of such an assumption might be that the underlying SCM that generates the data to be inducted from is linear (as in NOTEARS (Zheng et al., 2018))

equations are of the form $f(\mathrm{pa}_X, N_X) = \boldsymbol{w}^\mathsf{T} \mathrm{pa}_X + N_X$. For CGNN (Goudet et al., 2018), the authors perform a score-based approach i.e., generative neural networks are being fit to the data for each of the pausible DAG-structures that the SCM could take on and the best-performing one is subsequently being selected. Again, since SCM can be converted into CEM (Prop.2,Def.2), CGNN are interpretable via SCI. For DAG-GNN (Yu et al., 2019), the authors make the same assumption of linear SCM as in (Zheng et al., 2018) and adapt them to graph neural networks, $\boldsymbol{X} = f_2((\boldsymbol{I} - \boldsymbol{A}^T)^{-1} f_1(\boldsymbol{Z}))$ where $\boldsymbol{X}, \boldsymbol{A}, \boldsymbol{Z}$ are data, graph and normal noise respectively and $f_i$ the GNNs. For our final example, the NCM (Ke et al., 2019; Xia et al., 2021), the proof is trivial since the NCM model class is a subset of the space of all SCMs and therefore also CEM-estimator (Prop.2). $\qquad\square$

## A.5 INTUITION ON CONJECTURE 1

While the establishment of SCI (Thm.2) and the consequential insight that any neural induction method is thereby interpretable (Thm.3) stand as separate results, we further considered the question of whether one could in fact make use of pre-existing SCI for learning. While we don't provide a proof, we generally observe empirically that, if a human modeller or any other type of a-priori inductor provides some SCIs, then one can use these SCIs to improve learning. An explanation for the observed improvement might lie in the required consistency of underlying CEM (sub-property of the unobserved SCM). I.e., the model needs to infer a structure that can provide for consistent interpretations to subsequent why-questions aligned with the answers provided by the existing interpretations (be human or other). Therefore, we conjecture that this is generally the case as our evidence supports our belief that the statement is true, even if not proven. Formally, we state:

**Conjecture 1** (**SCI Regularization**). *Let $L$ be defined as in Def.3 and let $H^*$ be the true CEM. Regularizing $L_r := L + \frac{1}{n} \sum_i \|\mathcal{I}(Q_X, H, \boldsymbol{D}) - \boldsymbol{I}_i)\|_2^2$ with existing and overlapping SCI $\boldsymbol{I}$ leads to a more optimal solution $|\arg\min_{H \in \mathcal{H}} L_r(\boldsymbol{D}, H) - H^*| < |\arg\min_{H \in \mathcal{H}} L(\boldsymbol{D}, H) - H^*|$.*

## A.6 ADDITIONAL THEORETICAL RESULTS

The following statements are an addition to the theoretical results established in the main paper. In Prop.3 we state that any induction method is in fact a causal method, since there exist infinitely many possible SCMs that could act as the corresponding model. This view is inline with the SCM metric space theorem (Thm.1), making an (approximately) right choice of SCM important.

**Proposition 3** (**Induction Methods**). *Let $\mathcal{M}$ be a DAG-induction method for data-based inference, $\mathcal{M}(\boldsymbol{D}) = G$. Let $\mathcal{C}$ denote the space of all possible SCM. Then it holds that $\exists \mathfrak{C} \in \mathcal{C} : \mathfrak{C} \implies G$.*

*Proof.* A proof by construction is sufficient. Let $G = \{V_G, E_G\}$ be a DAG (variables, edges) as described in the proposition ($\mathcal{M}(\boldsymbol{D}) = G$). Then we can construct an SCM $\mathfrak{C}$ where the structural equations of $\mathfrak{C}$ use nodes $V_G$ to mimic $E_G$. Naturally, there will be infinitely many such $\mathfrak{C}$ since parametric form and parameterization can be arbitrary. However, the proposition suggests for the existence of *at least one* such SCM for any graph, and we are done due to $G$ being arbitrary. $\qquad\square$

Another important observation in Prop.4 follows from the SCI's computation arguments (Thm.2) being the CEM and not SCM (where we have shown that SCM are a more expressive model, Prop.2). I.e., the function which converts any SCM to their corresponding CEM as defined in Def.2 will not allow for two SCMs to coincide, that is, the function is injective (each mapping to CEM is unique).

**Proposition 4** (**Injective CEM Mapping**). *The function that maps SCM to CEM is injective.*

*Proof.* We assume the general case where the $f_{ij}$ components of the CEM are a causal effect from $i$ to $j$. Let $F : \mathcal{C} \mapsto \mathcal{M}$ denote said function that maps from the set of all possible SCMs to the set of all CEMs. Since CEM is a sub-structure of an SCM by definition (Def.2), there is a restriction on $F$ that each edge $i \to j$ of a given SCM is being bijectively mapped to a causal effect $f_{ij}$ for the CEM. Further, $F$ is restricted to have model each of the structural equations. As Thm.1 suggests, one can always find two SCMs of same size $\mathfrak{C}_1, \mathfrak{C}_2 \in \mathcal{C}_n$ for which will hold $d(\mathfrak{C}_1, \mathfrak{C}_2) > 0$, where $d_1$ is the measure from Thm.1, and thereby imply that $d_2(F(\mathfrak{C}_1), F(\mathfrak{C}_2)) > 0$ where $d_2$ is a matrix norm. $\qquad\square$

### A.7 ELABORATION ON SCI PROPERTIES

The three basic logic rules (Prop.1) dictate how the SCI (2) will look like for some causal estimate of the system and any given query and data. In our implementation, we additionally provide the actual relation $R$ as a return argument of each of the rules. This allows for a more fine-grained interpretation that gives insights on the specific relation to the current explaining reason. In a nutshell, it allows to extend a statement 'Y because of X' to a more detailed one like 'Y because of X being low'. The general pronunciation scheme for the the three rules (excitation, inhibition, and preference), that allows for a human-understandable natural language version of the SCI, are as follows:

| R1 | Excitation | 'Y because of X [being low/high]' |
| R2 | Inhibition | 'Y although X [is low/high]' |
| R3 | Preference | 'mostly' + R1 or R2 pronunciation |

Table 3: **Pronunciation Scheme.** Right shows the natural language reading of a rule's activation.

The pronunciation of the details to the relation e.g. 'low'/'high' is context-dependent in that these words might need to replaced with adequate/corresponding words suitable for the context i.e., 'the Matterhorn is cold because of the high altitude' ('cold Temperature because of Altitude being high') is fine while 'the remaining car fuel is low because of the bad driving style' ('low Fuel because of Driving Style being bad') requires the context-adaptation ('low' $\mapsto$ 'bad'). Another noteworthy detail to the SCI properties is the property of *non-repeating causes within interpretations* which reduces redundancy. Consider for instance our lead example on Hans's mobility (Int.1 or Fig.2), the CEM suggests that $F$ can also be explained by $A$, since $A \rightarrow F$. However, the corresponding SCI does not give this reason because of the aforementioned property which ensures that redundancy is being avoided. I.e., in the interpretation step before we actually explain $H$ using both $A$ and $F$, since $\{A, F\} \rightarrow H$, therefore, making it irrelevant to the query for explaining the relation between the parents $(A, F)$. While we provided intuition on the derivation of the basic rules (Prop.1) while alongside also providing a lead example (1,Fig.2), we now additionally motivate the namings 'excitation', 'inhibition' and 'preference' i.e., why we think they make sense. We took inspiration from neuroscience, where the former two terms relate to the way neurons interface with each other using their synaptic-dendric connections. The last term is a term to propose relativeness and thus a preference for one cause of over the other. All terms thereby adequately describe any causal path in qualitative terms while also providing an almost synonym-quality to the pronunciations (Tab.3).

### A.8 EXP.1: ASKING NEURAL MODELS TO INTERPRET THE LEARNT (THM.3)

We select NOTEARS (Zheng et al., 2018) as a representative data-driven induction method for the illustration in Tab.1 which considers the data sets and single-why queries illustrated in Fig.2 i.e., Deutscher Wetterdienst (DW, Mooij et al. (2016)), Causal Health (CH, Zečević et al. (2021a)), Mileage (M), and Recovery (R, Charig et al. (1986)). Fig.5 additionally visualizes the generated SCI that are in support of Thm.3 i.e., the interpretability of NIM. The SCI generated using the learned causal semantics are identical for the DW and M data sets, while differing only subtle for R and drastically for CH data sets. The former discrepancy occurs on the second-level of reasoning i.e., the right top-level explaining answer is given to the question (i.e., "Kurt did not recover because of the problematic pre-conditions") but was contrasted wrongly (i.e., the treatment countering the state of condition and not affecting the condition). The latter discrepancy revolves around a totally different structure e.g. the learned model expects a direct cause-effect relation between age and mobility while also wrongly assuming that nutrition has a detrimental effect on health. An explanation in the case of NOTEARS is clearly the violation of the linearity assumption for the CH data set.

While in Thm.3 we prove that NIM are generally interpretable in the sense of SCI (Thm.2), for empirical illustration we also provide more examples of such NIM-based SCI, as we did with our lead examples for NT, in this case additionally for CGNN (Goudet et al., 2018) and DAG-GNN (Yu et al., 2019). Tab.4 shows an application of these methods to a superset of questions (that is, same and more) and same data we used for NT. It is crucial to note that the presented results have *not* been hyperparameter-optimized (HO). Take for example CGNN, where candidate selection is exhaustive (brute force, and thus super-exponential in the number of nodes) and the model selection heavily relies on the neural approximation, thereby, HO is likely to be important. In a nutshell, the motivation behind Tab.4 is to present support for our theoretical proof on SCI-interpretability of NIM

methods i.e., we also give empirical proof for several methods in practice (opposed to pure theory). To assess the quality of the SCI, it is important to note the assumptions made by the original method. E.g., NT and DAG-GNN assume linear SCM. Thereby, we have no guarantees for running such a method in a non-linear data domain (which we do with the data sets DW and CHD). Interestingly, these assumptions can in fact be exposed by SCI. Consider the DW data set (Tab.4, first example), theory suggests that a linear model with Gaussian noise will exist in both directions $X \rightarrow Y$ and $Y \rightarrow X$, thus being non-identifiable (Peters et al., 2017). Methods like NT and DAG-GNN therefore pose the assumption that the given data comes, in this case, from a linear model with Gaussian noise i.e., the identifiability problem is being circumvented altogether. This is also the reason why different random seeds can lead to both modellings ($A \rightarrow T$ and $T \rightarrow A$) for the DW data set (see in Tab.4 how the SCI flips for $\mathcal{M}_3$=DAG-GNN for the two opposing DW queries). Another important note is that the uninformed SCIs "No causal interpretation ..." occur when the method's CEM estimate does not contain a causal path to the variable that is being queried i.e., the CEM will actually contain a non-trivial estimate of the underlying causal structure, even though the SCI returns a trivial/empty interpretation since the variable of interest can not be reached within the estimate's structure with a directed path (i.e., the base case in Thm.2 is trivially triggered). In fact, these negative 'no answer'-type of cases are important since the model need also be able to know when there is nothing to be known. For this case, we also pose why-questions to which the ground truth is already a 'no answer' interpretation since there is no causal connection to the variable being queried by the why-question. The empirics in Tab.4 suggest, as theoretically proven (Thm.3), that the NIM are interpretable and also that all 3 rules (excitation, inhibition and preference) are being used for the NIM-based SCI. As a positive example, consider example #3 for the CH data set where $\mathcal{M}_1$ captures the complex interpretation correctly up to preference and falsely assuming that nutrition ($F$) has a negative causal effect on health ($H$). A more interesting example (#8 for the R data set) shows that the main reason being bad pre-conditions ($Z$) is being captured but the model falsely assumes that those are because of the received treatment ($T$). To consider a negative example have a look at example #4 again for CH where the actual answer is a 'no causal interpretation' since age ($A$) is a root node. However, $\mathcal{M}_3$ claims that the age is high because of the nutrition and mobility ($M$), then again because of health. While the statement is wrong and also feels exaggerated, inspecting closely one can detect the correct existence of the causal edge between mobility and health ($H \rightarrow M$). I.e., the model interprets wrongly, but its causal model is still partially valid.

## A.9 EXP.2: USING INTERPRETATIONS TO IMPROVE LEARNING (CONJ.1)

We again make use of the NOTEARS (Zheng et al., 2018) method as representative of graph induction methods for the subsequent experiment in which investigate Conj.1 i.e., whether existing interpretations can be used as a supervision signal to improve the quality of induction. To circumvent the non-differentiable nature of our recursive interpreter from Thm.2 we train a neural network on a set of legal interpretations to mimic the interpreter while being fully differentiable. Mathematically, we set $\mathcal{I} = \mathfrak{I}_\psi$ in Alg.1 where $\mathfrak{I}_\psi$ is said neural approximation to the SCI recursion from Eq.3. Following (Zheng et al., 2018), we generate 70 random linear causal models following Erdos–Renyi structures. We use graph induction to infer 70 more graphs, making 140 in total. For each graph we generate 50 random single-why questions to be answered, resulting in a data set of 7000 interpretations. All the (discussed) details regarding the learning problem and the results are additionally being illustrated within Fig.6. We extend the NOTEARS loss composition with this neural approximation to the regularization we proposed in Conj.1 and perform graph induction once with and once without the regularization (where between 1 and 50 interpretations are being observed). The graph induction is being performed in a data-scarce setting with only 10 data samples per graph induction. Thus to infer the true causal structure the method ideally needs to perform sample-efficient. Fig.3 shows our empirical results on the error distributions for all the graphs while presenting the qualitative difference in the estimated graphs for the most significantly improved example. It can be observed that with the regularization the induction method can both identify more key structures while significantly reducing the number of false links, thereby appearing to be overall more sample-efficient. An explanation would be that, as conjectured, the interpretations contain valuable information about the underlying CEM if the interpretations themselves were generated by a similar CEM, thereby striking structures that would lead to contradicting interpretations.

A.10   Exp.3: Human Case Study, Algorithmic Versus Human Interpretations

We instructed $N = 22$ participants to answer our questionnaire (Fig.7). The questionnaire asked the following questions: Given a pair of variables, does a causal relationship exist (existence)? If yes, then which is the cause and which is the effect (direction)? If there are multiple causes for a single variable, then how impactful is each of the causes (preference)? All of these questions, alongside their responses, are of qualitative and subjective nature. It is important to note that the participants *do not* perform the actual induction from specific, provided data like the algorithms do i.e., the human subjects are not given the variable names nor concrete data points that would allow them to find the rules for the specific data sets. Instead, they were only given the variable names/depictions, thereby having to induct from personal experience/understanding essentially. This approach to human induction is related to the experimental setups in (Griffiths & Tenenbaum, 2006; Hattori, 2016).

The motivating lead research questions we intended to answer, and in fact do answer successfully with this experiment, are: What are CEM (Def.2) that (some) human could model? How does overlap for human-based CEM occur? How do subsequent SCI (Thm.2) between humans and algorithms differ? In a nutshell, we wanted to investigate the similarity of CEMs between subjects in addition to the similarity between subjects- and algorithm-based SCIs.

A caveat regarding the analysis and interpretation of human judgements is that sample bias may distort conclusions. Sample bias has long been identified within the behavioral and social sciences as limiting the generalization of results obtained in a specific sample to the population. A common methodological fix to counteract such biases is to increase the sample size, see (Daniel, 2017) for a recent application and discussion. Certainly, the observed sample will affect the way the difference (to e.g. algorithm-based SCI) turns out to be, but then again our research question is *not* concerned with all possible human interpretations, but any. Furthermore, we chose data sets that model very general examples and thus offer accessibility to the general population since no single person might be an expert. Ultimately, this way of designing our experiment, while not removing sample bias of course, renders the bias's qualitative effect onto our subsequent investigation negligible.

In the following we provide a discussion of several interesting and important insights discovered through the human user study. Nonetheless, it is important to note that our results like most modern day interpretations of human behavior are of conjectural nature – sensible, educated guesses essentially. During this discussion, we will point to specific aspects of the descriptive statistics displayed in Fig.8. The actual human data is also being appended for the sake of completion (Figs.9,10,11,12,13,14). The questionaire contains four examples with two, three, three, and four variables (or concepts) respectively that are being visually depicted in addition to a concise textual description. We randomized the textual description of up to three variables across all examples for any randomly selected participant. Doing so, we allow for the randomized concept to reverse causal influence directions, thus, diminishing the bias of chance-selecting said causal direction – in a nutshell, this randomization scheme helps us in controlling for interpretation variance (or leeway) of the subjects. Nonetheless, we still observed that for any variable pair $(X, Y)$ the meanings of $X$ and $Y$ themselves could be interpreted differently, which ultimately resulted in False Negatives regarding agreement i.e., people will disagree technically although they actually agree. To give a concrete example, consider the following: pre-condition in Example 2 can be interpreted as "the length of the medical history of a patient" (negative; increasing implies lower chance of recovery) opposed to "the state of well-being of a patient" (positive; increasing implies higher chance of recovery), thereby some subjects might choose $Z_1 \rightarrow R$ while others will choose $Z_2 \leftarrow R$ where $Z_i$ are the different interpretations of the "pre-condition" concept (and $R$ denotes recovery), yet all subjects agree on an existing relation between the two variables: $Z_i \leftrightarrow R$. Also, some variables/concepts were more stable in their interpretation variance. To give yet another specific example, altitude and temperature in Example 1 (Fig.7) are stable concepts while the aforementioned pre-condition in Example 2 is unstable (due to its interpretation variance/leeway). More importantly these different interpretations due to the ambiguity inherent in language become visible within the statistics. To stay inline with the previous example, consider the medical example (E2) within Fig.8 (second row, middle) and specifically consider the edges $T \rightarrow R$ and $Z \rightarrow R$. For the former relation the agreement between subjects is evident i.e., the majority of human subjects will select this edge. For the latter relation, we clearly see the two previously discussed interpretations that subjects employ during edge decision. I.e., for some subjects the edge between $Z$ and $R$ is positive and for some

others it is negative, while naturally all agree upon there being a relation between the variable pair ($Z \leftrightarrow R$) opposed to there being no relation ($Z \nleftrightarrow R$).

We observe a systematic approach and thereby non-random approach to edge-/structure-selection by the human operators, see any of the subplots within Fig.8. Furthermore, there are only a few clusters even with increasing hypothesis space. Both the systematic manner and the tendency to common ground are evidence in support of the CMMC hypothesis (Hyp.1) and the argument on true causality to be more likely within the overlap of SCMs (Thm.1).

Although we randomize the order of variables in addition to consistently presenting them in a simple line with the intention of not inducing any specific sorting/structure to avoid bias, we still observed apparent, unintended subject behavior. For instance, subject number 5 (Fig.10) only considered pairs presented next to each other as being questioned although the other combinations are meant to be queried as well. While additional research needs to corroborate these observations, our data suggests that attention might have decreased over the course of the experiment for a subset of subjects as suggested by e.g. subject number 7 (Fig.10) where overall agreement with the subject majority is to be found but eventually at the very last example 'mistakes' occur (specifically, the subject highlighted that 'increasing age increases mobility', in stark disagreement with the majority of participants). We also observe that the increase in hypothesis/search space (i.e., more variables) comes with an increase in variance. TThis variance increase can be argued to be due to the progressive difficulty of inference problems as well as decreased levels of attention and potential fatigue across the duration of the experiment (e.g. consider the duplicate plots, third column, in Fig.8 where the number of unique structures that are being identified increases significantly). Yet another interesting observation concerns the aspect of time, consider subject number 17 (Fig.13) where there is a cycle between treatment and recovery where the subject likely thought in terms of 'increasing treatment increases speed of recovery *which subsequently* feeds back into a decrease of treatment (since the individual is better off than before)' which seems like a valid inference but clearly considers the arrow of time. Yet another observation, some subjects faced questions of variable scope e.g. if there is a causal connection between nutrition and mobility, then some subjects considered energy as the mediator and since energy is not part of the variable scope, confusion might arise whether to place an edge between nutrition and mobility or not. In fact, for such a scenario the correct answer is to place an edge, since there exists a causal path from nutrition to mobility, via energy, even if energy is not displayed. I.e., in causality, an edge can/will talk implicitly about all the more fine-grained variables that are part of the causal edge/path.

The second data set (E2) is an instance of the famous Kidney Stonde example (Peters et al., 2017), where $Z$ is a confounder that indicates the pre-conditions in terms of e.g. the size of the kidney stone, and it also illustrates the famous Simpson paradox (Simpson, 1951; Pearl, 2009; Peters et al., 2017) where the recovery will favor one treatment in the overall statistics while being better for all of the non-consolidated views for the other treatment. We observe that not a single subject places the edge pre-condition to treatment ($Z \rightarrow T$) which is arguably at the core of Simpson's paradox. This observation gives an additional cue on why the phenomenon is called paradox because no human subject expects the existence of this connection and even actively neglect the existence.

**Technical Details and Code.** All experiments are being performed on a MacBook Pro (13-inch, 2020, Four Thunderbolt 3 ports) laptop running a 2,3 GHz Quad-Core Intel Core i7 CPU with a 16 GB 3733 MHz LPDDR4X RAM on time scales ranging from a few seconds (e.g. evaluating SCI in Exp.1) up to approximately an hour (e.g. SCI-based learning in Exp.2). Our code is available at: `https://anonymous.4open.science/r/Structural-Causal-Interpretation-Theorem-D5C0`.

**Full-width Material.** Following this page are the figures and tables (Fig.5,6,7,8,9...14,Tab.4) that were referenced in the corresponding sections of the appendix.

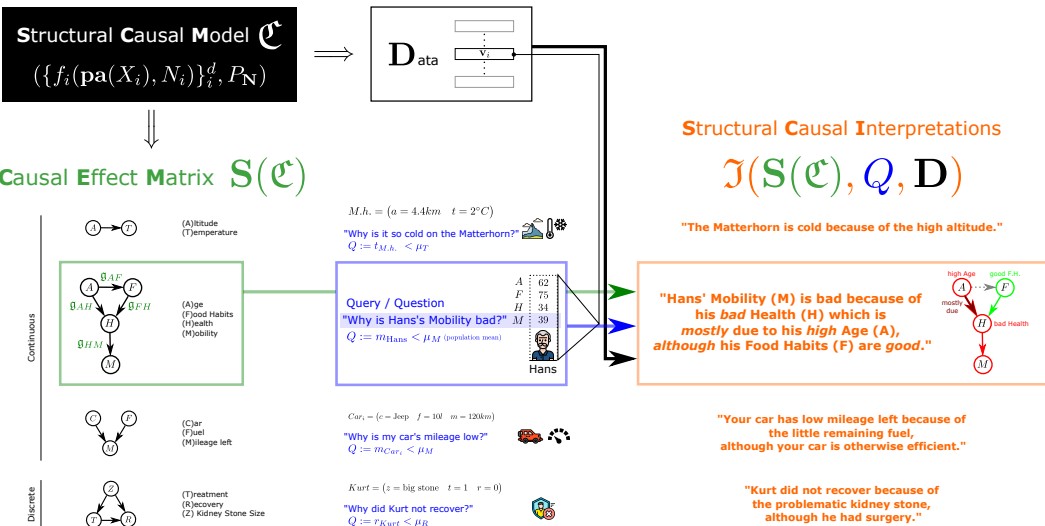

Figure 5: **Extension to Fig.2** In addition to the previous example, we add three more SCMs, both continuous and discrete alongside their why-questions $Q$ and resulting Structural Causal Interpretations $\mathfrak{I}(\boldsymbol{S}(\mathfrak{C}), Q, \boldsymbol{D})$. Our lead example concerning Hans's mobility is being highlighted alongside the intuitive-level computation. (Best viewed in color.)

| | Structural Causal Interpretation (Thm.2) |
|---|---|
| #1 | Dataset: DW, Query: "Why is the temperature at the Matterhorn low?" |
| GT | "The temperature at the Matterhorn is low because of the high altitude." |
| $\mathcal{M}_1$ | "The temperature at the Matterhorn is low because of the high altitude." |
| $\mathcal{M}_2$ | "The temperature at the Matterhorn is low because of the high altitude." |
| $\mathcal{M}_3$ | "No causal interpretation for Matterhorn's temperature." |
| #2 | Dataset: DW, Query: "Why is the Matterhorn so high?" |
| GT | "No causal interpretation for Matterhorn's altitude." |
| $\mathcal{M}_1$ | "No causal interpretation for Matterhorn's altitude." |
| $\mathcal{M}_2$ | "No causal interpretation for Matterhorn's altitude." |
| $\mathcal{M}_3$ | "The altitude of the Matterhorn is high because of the low temperature. |
| #3 | Dataset: CH, Query: "Why is Hans's mobility bad?" |
| GT | "Hans's mobility is bad because of his bad health which is mostly due to his high age, although his nutrition is good." |
| $\mathcal{M}_1$ | "Hans's mobility is bad because of his bad health which is bad because of high age and mostly due to his good nutrition." |
| $\mathcal{M}_2$ | "Hans's mobility is bad because of his good nutrition." |
| $\mathcal{M}_3$ | "No causal interpretation for Hans's bad mobility." |
| #4 | Dataset: CH, Query: "Why is Hans old?" |
| GT | "No causal interpretation for Hans being old." |
| $\mathcal{M}_1$ | "No causal interpretation for Hans being old." |
| $\mathcal{M}_2$ | "No causal interpretation for Hans being old." |
| $\mathcal{M}_3$ | "Hans is old because of his good nutrition and bad mobility, which is because of his bad health." |
| #5 | Dataset: CH, Query: "Why is Hans's nutrition good?" |
| GT | "Hans's nutrition is good because of being older." |
| $\mathcal{M}_1$ | "Hans's nutrition is good because of being older." |
| $\mathcal{M}_2$ | "No causal interpretation for Hans's nutrition." |
| $\mathcal{M}_3$ | "Hans's nutrition is good because of his bad health and mobility." |
| #6 | Dataset: M, Query: "Why is your personal car's left mileage low?" |
| GT | "Your left mileage is low because of your small car and your bad driving style." |
| $\mathcal{M}_1$ | "Your left mileage is low mostly because of your small car and because of your bad driving style." |
| $\mathcal{M}_2$ | "No causal interpretation for the left mileage." |
| $\mathcal{M}_3$ | "Your left mileage is low because of your small car and your bad driving style." |
| #7 | Dataset: M, Query: "Why is your personal car small?" |
| GT | "No causal interpretation for the car size." |
| $\mathcal{M}_1$ | "No causal interpretation for the car size." |
| $\mathcal{M}_2$ | "Your personal car's size is small because of your good driving style and fuel savings." |
| $\mathcal{M}_3$ | "No causal interpretation for the car size." |
| #8 | Dataset: R, Query: "Why did Kurt not recover?" |
| GT | "Kurt did mostly not recover because of his bad pre-conditions, although he got treatment." |
| $\mathcal{M}_1$ | "Kurt did not recover because of his bad pre-conditions which is because of the treatment he got." |
| $\mathcal{M}_2$ | "No causal interpretation for Kurt's recovery." |
| $\mathcal{M}_3$ | "No causal interpretation for Kurt's recovery." |
| #9 | Dataset: R, Query: "Why did Kurt get treatment?" |
| GT | "Kurt got treatment because of his bad pre-conditions." |
| $\mathcal{M}_1$ | "No causal interpretation for Kurt's received treatment." |
| $\mathcal{M}_2$ | "Kurt got treatment because of his bad pre-conditions." |
| $\mathcal{M}_3$ | "No causal interpretation for Kurt's received treatment." |

Table 4: **More NIM-based SCI.** We prove Thm.3 for general NIM while pointing to some example methods from the existing literature on NIM. Here we show the results of running the methods $\mathcal{M}_i$, 1:NT (Zheng et al., 2018), 2:CGNN (Goudet et al., 2018), 3:DAG-GNN (Yu et al., 2019) on the four data sets Deutscher Wetterdienst (DW, Mooij et al. (2016)), Causal Health (CH, Zečević et al. (2021a)), Mileage (M), and Recovery (R, Charig et al. (1986)) for the respective queries. As suggested, the methods are interpretable and reveal insights onto the learned causal semantics, while varying signficantly in quality in terms of accuracy relative to the ground truth (GT). Independent of accuracy, "No causal interpretation ..." occur when the CEM estimate of $\mathcal{M}_i$ contains no causal path to the queried variable $X$ i.e., $\mathrm{pa}_X = \emptyset$ (supported through GT sparsity). We also show GT interpretations that require a negative 'no answer' response by $\mathcal{M}_i$. The corresponding appendix sections cover a detailed elaboration for these NIM-based SCI (Thm.2).

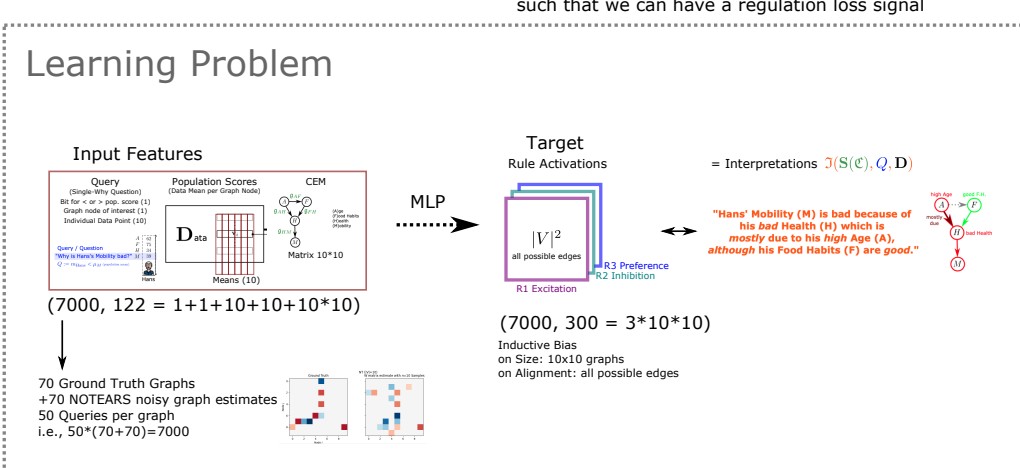

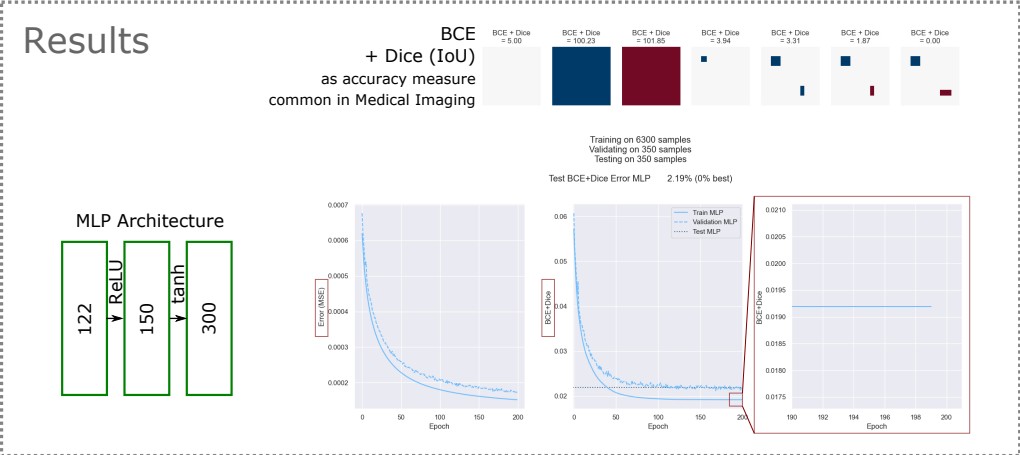

Figure 6: **Differentiable Interpreter.** To cope with the probelm of non-differentiability of the SCI Thm.2 we propose a learning problem in which we learn a MLP to mimic a set of existing legal interpretations. The neural network thus approximates the construction of SCI while being fully differentiable. (Best viewed in color.)

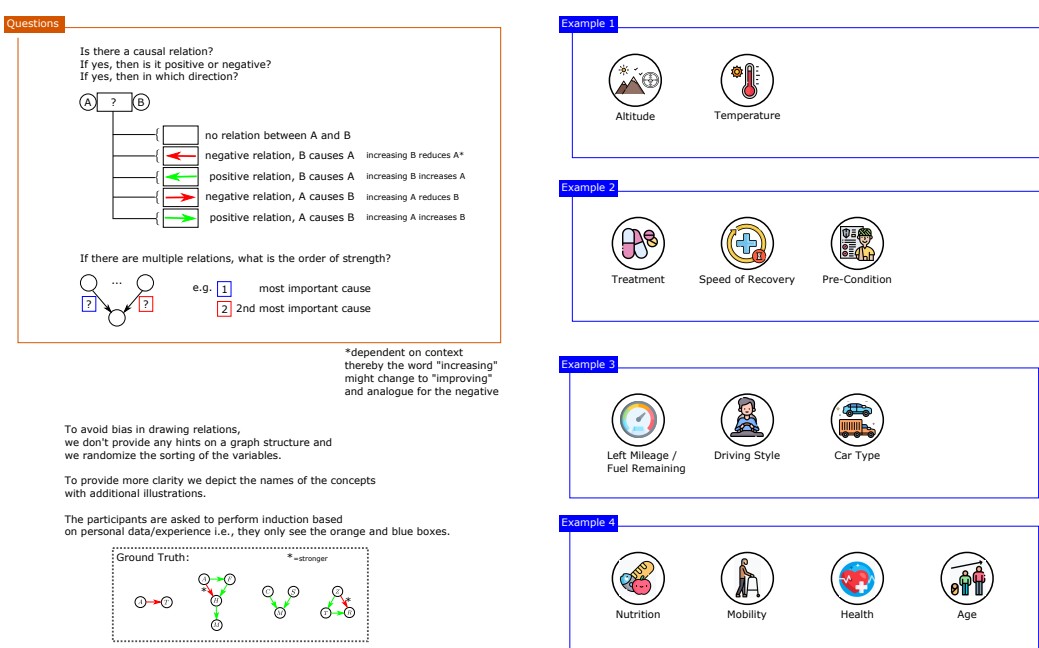

Figure 7: **Experiment Setup for the Human Case Study.** The participants are being asked two questions: whether there is a directed relation between some variable pair $A$ and $B$, and when there are multiple causes how they behave relatively i.e., the order of strength in relations. We avoid bias in drawing relations by randomizing the order and presenting the variables in a sequence. Induction is being performed from personal 'data'/experience, rather than by looking at a matrix of data points. (Best viewed in color.)

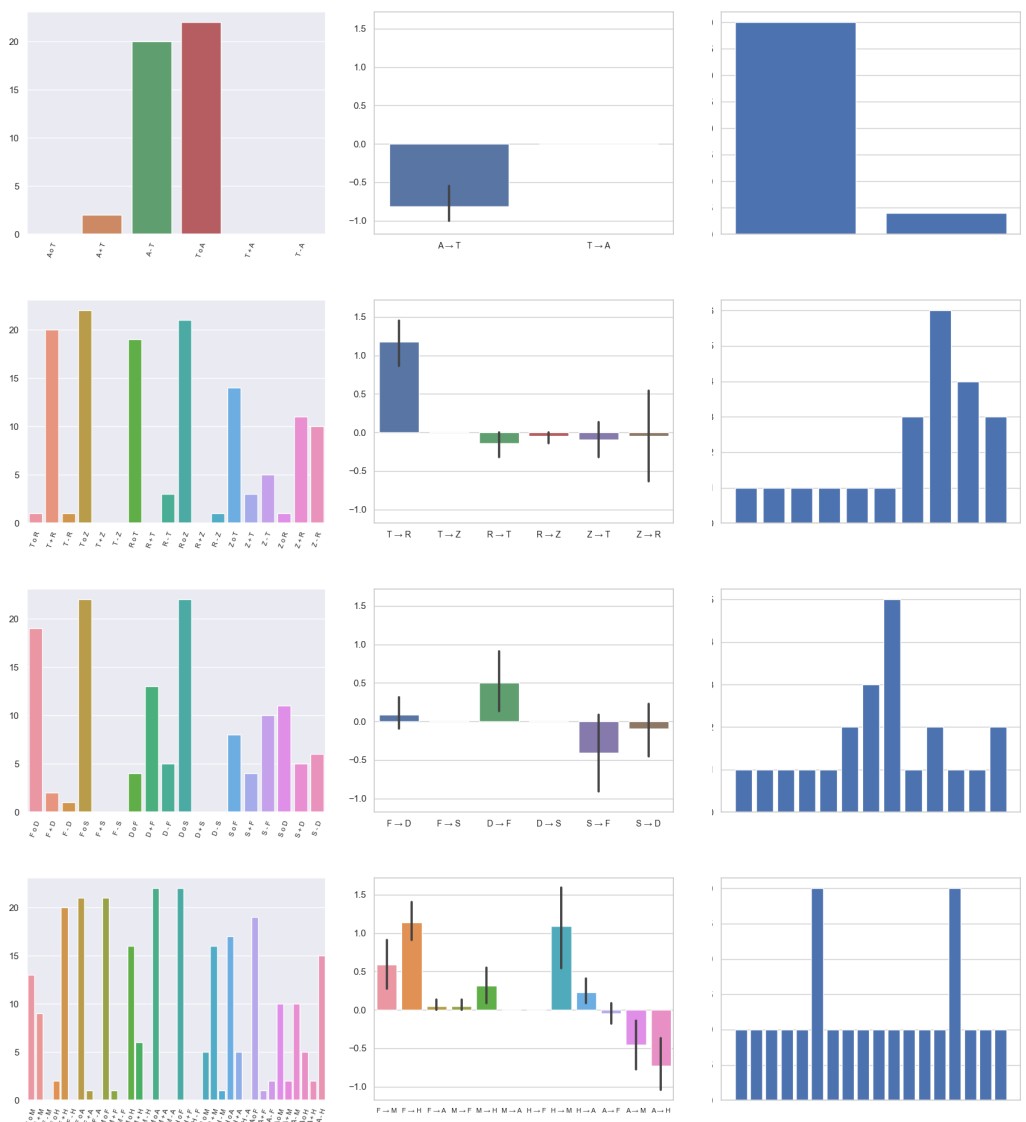

Figure 8: **Human Data Analysis: Qualitative, Quantitative, and Uniqueness.** Statistics collected from the human data ($N = 22$). The rows denote the four data sets: Deutscher Wetterdienst (DW, Mooij et al. (2016)), Causal Health (CH, Zečević et al. (2021a)), Mileage (M), and Recovery (R, Charig et al. (1986)). The columns: qualitative edge distributions that show for each of the different edge type how often it was chosen respectively (left), quantitative edge distribution for each edge where the error bars denote confidence intervals (middle), and the unique structure counts where each bar depicts the frequency of a qualitative structure discovered by the human subjects (right). Extensive elaboration on the setup, execution and results of this human study are to be found in the corresponding appendix section. (Best viewed in color.)

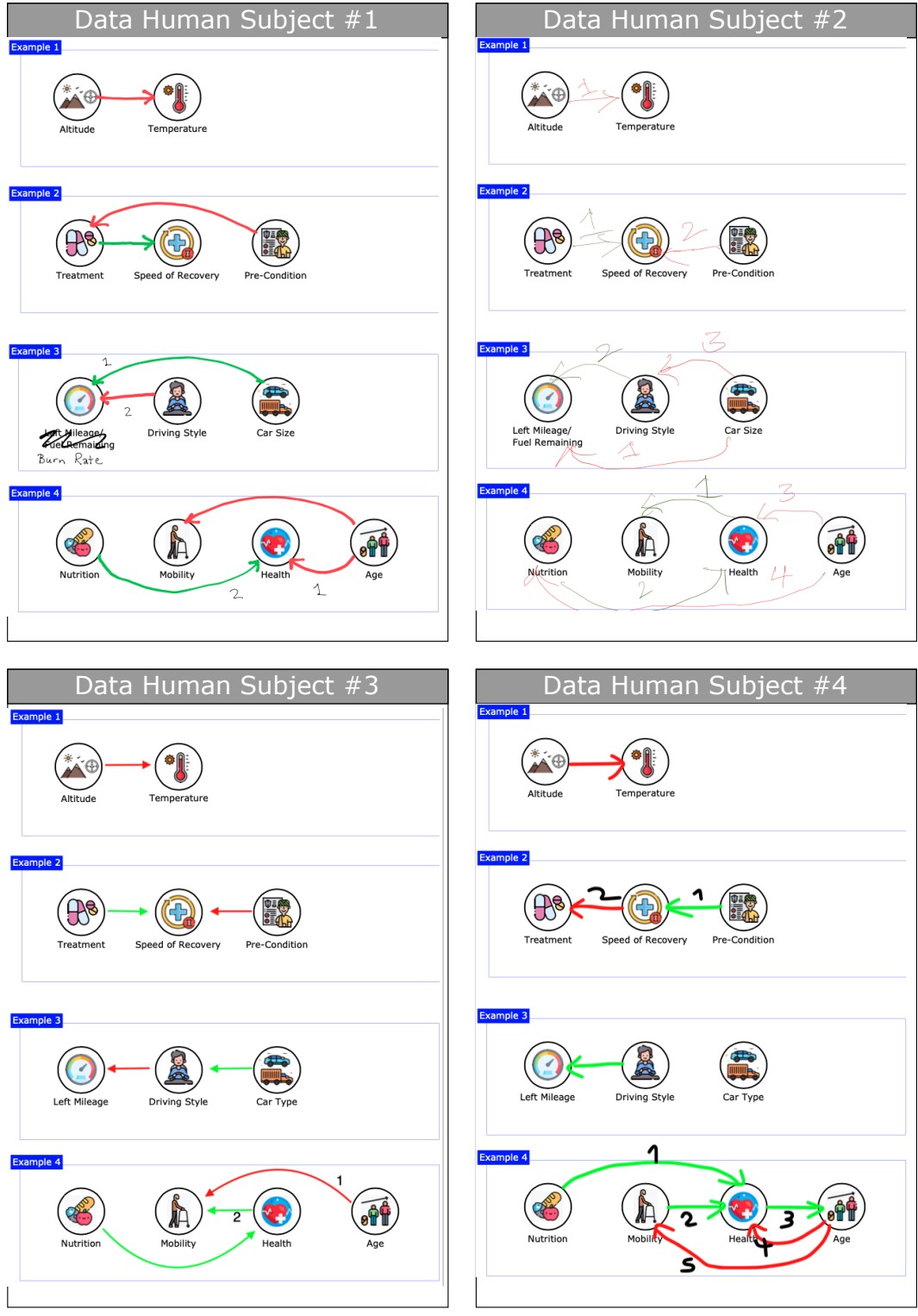

Figure 9: **Actual Human Data Anonymized.** Subjects 1-4. Answers followed the questionnaire in Fig.7. (Best viewed in color.)

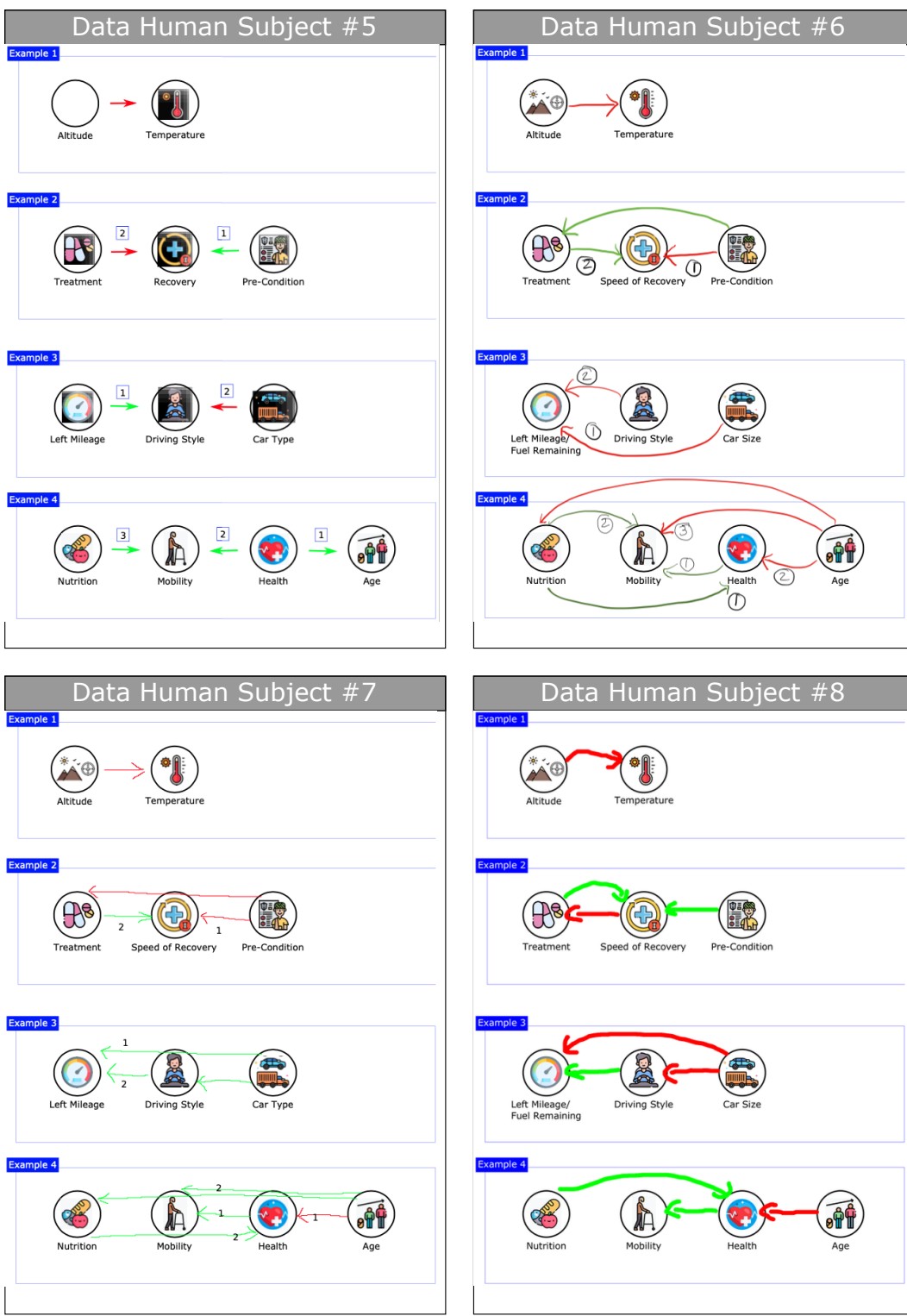

Figure 10: **Actual Human Data Anonymized.** Subjects 5-8. Answers followed the questionnaire in Fig.7. (Best viewed in color.)

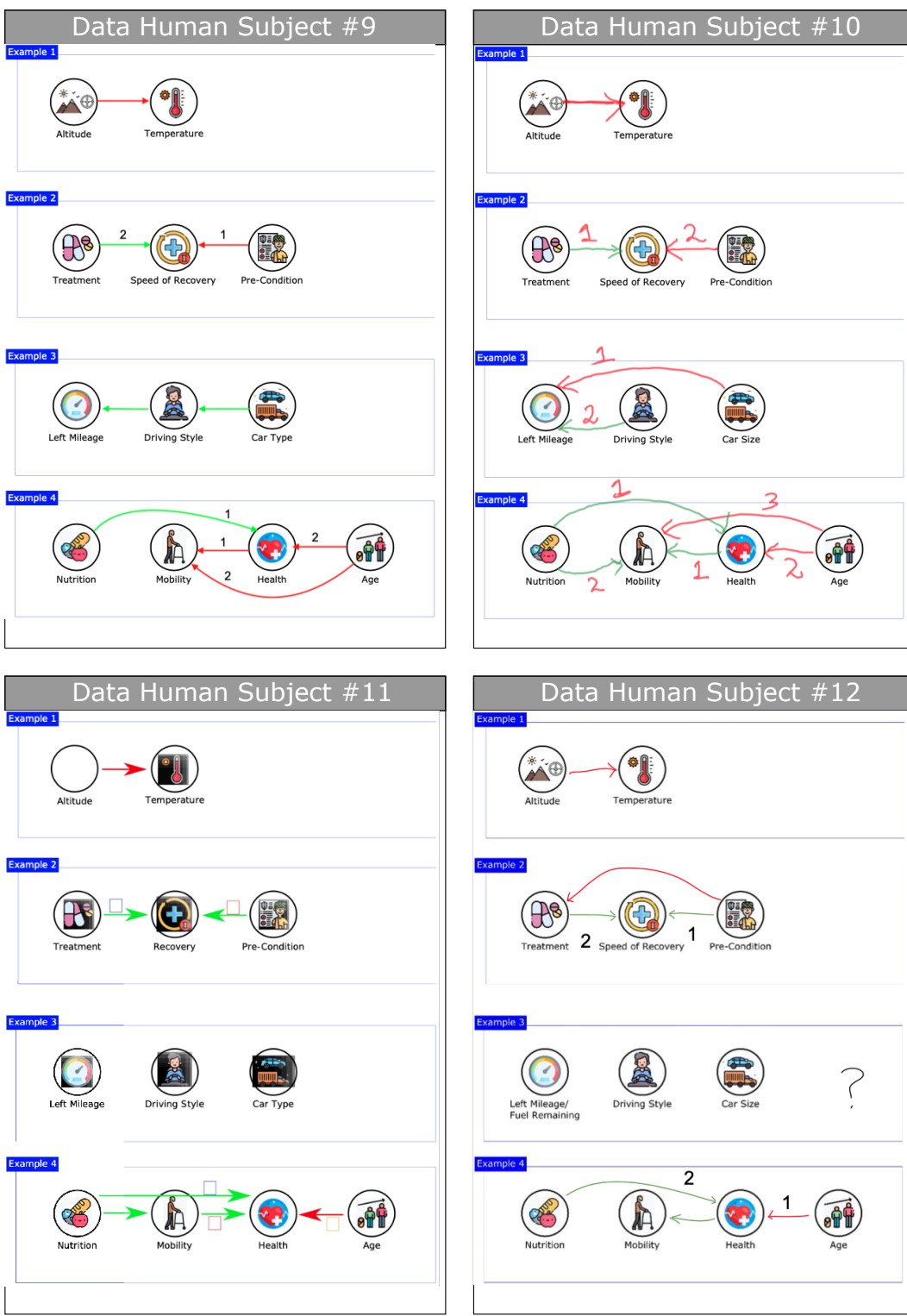

Figure 11: **Actual Human Data Anonymized.** Subjects 9-12. Answers followed the questionnaire in Fig.7. (Best viewed in color.)

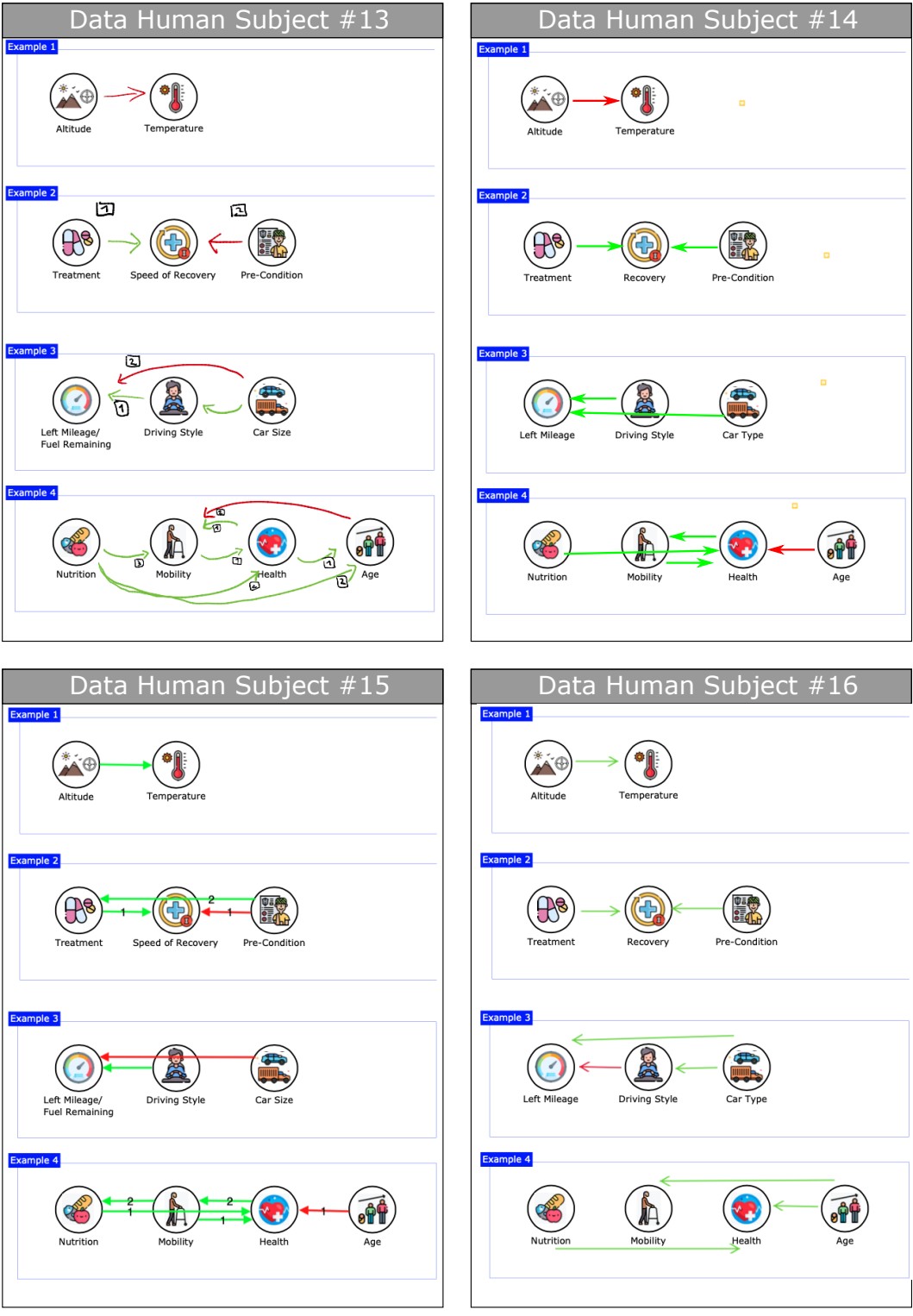

Figure 12: **Actual Human Data Anonymized.** Subjects 13-16. Answers followed the questionnaire in Fig.7. (Best viewed in color.)

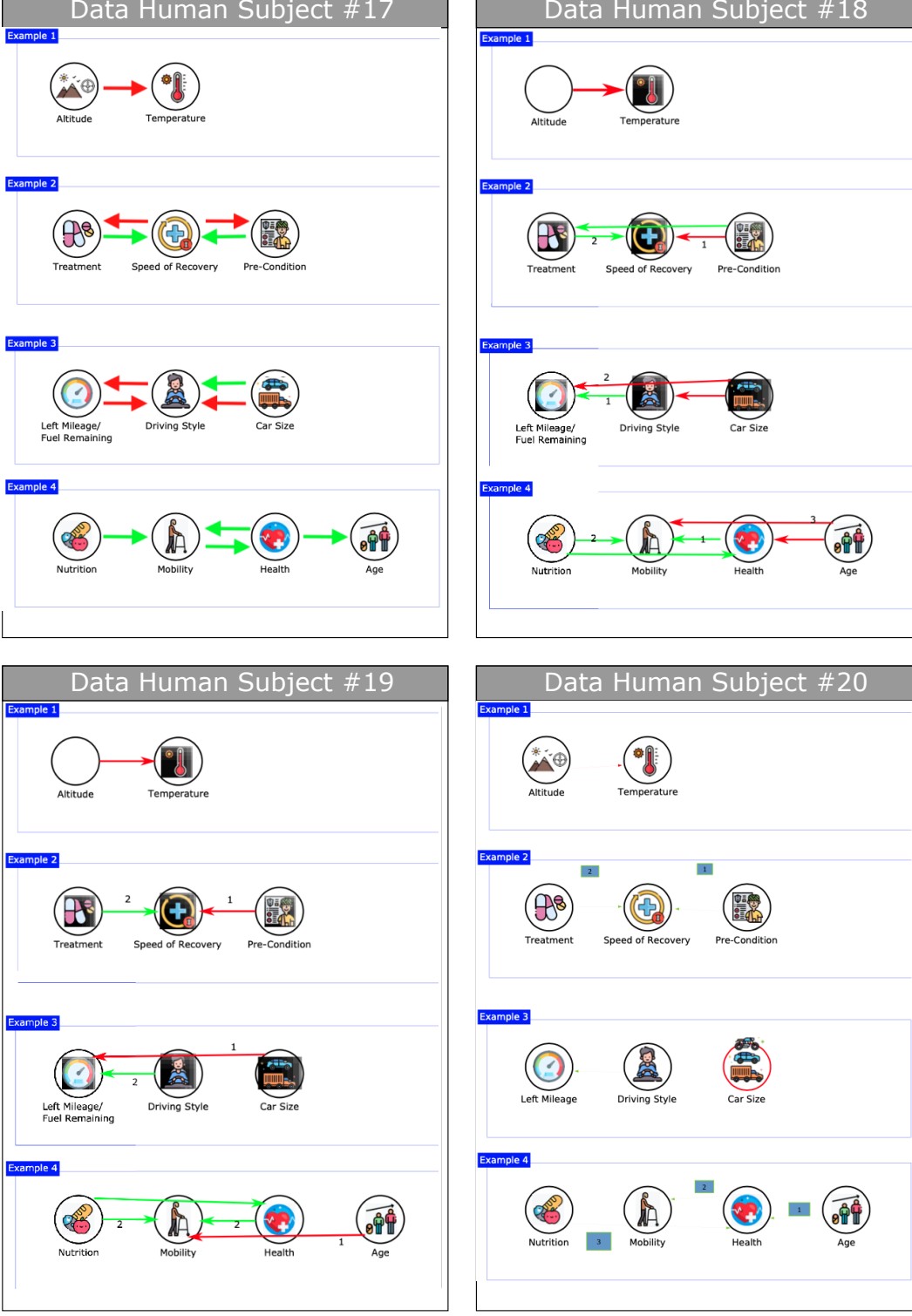

Figure 13: **Actual Human Data Anonymized.** Subjects 17-20. Answers followed the questionnaire in Fig.7. (Best viewed in color.)

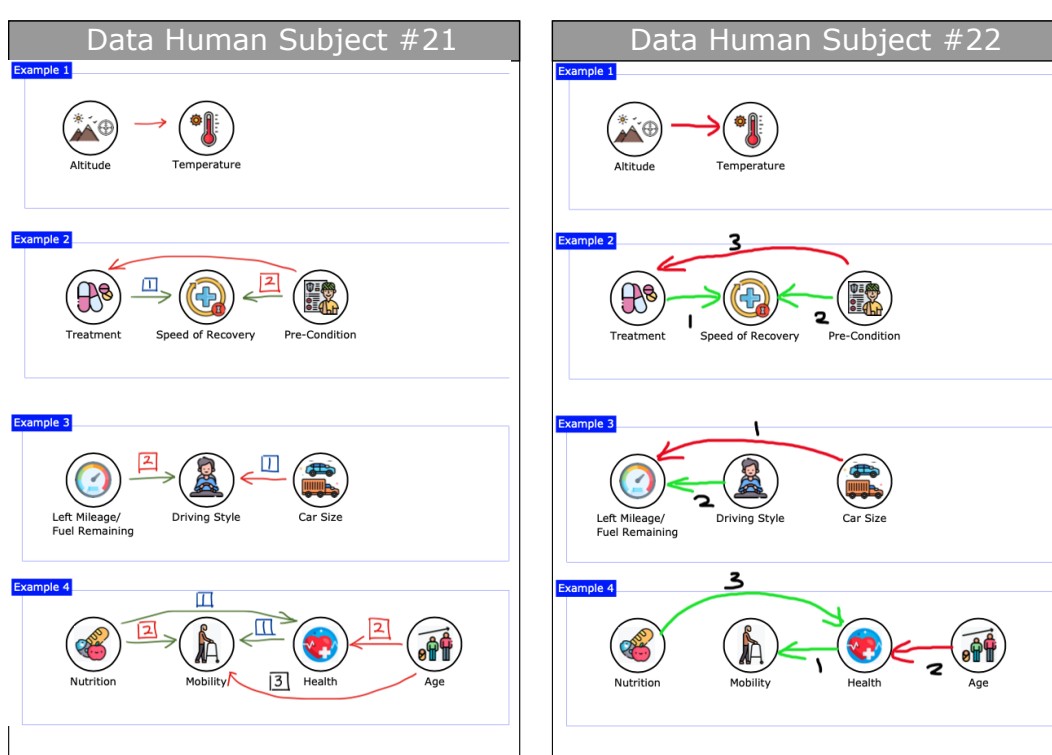

Figure 14: **Actual Human Data Anonymized.** Subjects 21 and 22. Answers followed the questionnaire in Fig.7. (Best viewed in color.)

