# OpenReview forum: "Structural Causal Interpretation Theorem"
_ICLR.cc/2022/Conference — ICLR 2022 Submitted_

### Official Review · Reviewer_2Fy5 · 2021-10-31

**Correctness:** 3
**Technical Novelty And Significance:** 3
**Empirical Novelty And Significance:** 4
**Recommendation:** 8
**Confidence:** 4

**Main Review:**

This paper combines causality, interpretation, and machine learning in an intriguing way.

Let me first briefly go over the paper in order. Section 3.1 introduces a hypothesis (CMMC) that mental models can be represented as an SCM and Theorem 1 talks about metric space.  Section 3.2 provides an example which overviews data-driven interpretation of why a variable attains a value. Then, Section 3.3 formalizes the notion of why question, first-order logic for interpretations, and causal effect matrix. These definitions lead to SCI theorem, and the authors further made a connection between Neural Induction Models and SCI-interpretability.

To begin with, the paper’s strong assumptions are not quite well articulated.
While the causal model of reality can be very sophisticated, the assumed SCM is quite simple. In page 2, “product distribution over noise variables” and “independent, exogenous variables” imply that the SCM yields a Markovian model, meaning that we observe all the confounders among the measured variables. This is not typically assumed in causal inference literature where causal effect is trivially identified from observational data. Further, the Theorem 1’s definition of structural equation that f_i can be decomposed into an additive equation with no interaction among parents is very strong. This definition is lurking inside theorem as “i.e.” while this assumption is crucial in understanding the precise condition the later theoretical developments rely on and the limitations of the proposed method. Interpretation provided by edge-specific causal effects can be problematic in most situations (without the additivity assumption). If Y has two parents A and B, it may be possible to see that both A and B have positive effects on Y but they can have jointly a negative effect on Y. The author might want to consider such a combinatorial nature in SCI.

Next, I find the wordings in the paper is a bit broad (“why question”, “interpretation”) which may confuse readers, and some of the definitions (what I perceived) are presented as proposition and theorem.
The definition of a single why question seems reasonable but the name doesn’t seem to be specific enough to properly represent what it really is about. Is every why question about the inequality based binary relationship? Similarly, the name for Proposition 1 "First-Order Logic Rules for Interpretation“ seems a bit broad. Could you please provide any reason why it is a proposition not a definition. (what were claimed and proved?) The authors mentioned that “the three rules in Prop.1 are naturally derived from the principles established in the previous section.“ It would be very helpful if such principles were explicitly mentioned, and how the rules are naturally derived. SCI theorem is recursively applying the three rules in proposition 1 and concatenating the result. Similar to Prop 1, Is there any reason why it is a theorem rather than a definition for what the authors want to do with DAG in a specific way?

I would like to point out a few more things with no particular order.

Where do you use the defined metric space?

In the middle of page 3. I couldn't fully follow the logic that “Interestingly, this comparability of SCM …” leads to “This insight tells us …”.

Given that the model is Markovian, P(Vi|Pa_i\V_j, V_j)=P(Vi|Pa_i\V_j, do(V_j)). Then the causal effect in the causal effect matrix is sum_{rest of parents } P(V_i | Pa_i\V_j, V_j) P(Pa_i\V_j). Hence, the causal effect can be driven from data (if you have a causal graph). Is it correct? Again, as mentioned earlier, it seems that just using this single-variable specific "aggregated" causal effect seems too restrictive.

Eq (3) is interesting since explanation (interpretation) often involves “back-tracking”! If we consider the interaction among parents, better back-tracking would be possible.

Def 3 “Family of CEM Estimators“ seems to be one of the crucial definition but the explanation is lacking. You have a (typical) loss function (e.g., negative log likelihood) how good a “causal effect labeled DAG” is. What do you mean by “The family of CEM estimators encompasses any learner …”?

Why does Theorem 3 exclude any other conditional independence based or score-based causal discovery algorithm which would generate interpretable DAGs (up to its Markov equivalence in some cases) but only NIM-DAGs? Is this because this paper has very restricted assumptions on what an underlying SCM is (linear and Markovian)?

The algorithm 1 needs to be edited.
Input: Can you share some concrete examples of bold I?
Line 2, “while” is “for each I_i in I ? (since no i++)
Line 4, how I_i itself returns two objects without calling any function on it?
Line 5 returning I hat?
Line 6 dangling closing parenthesis?
Line 7. Is the updated parameter reused in Line 3 within the loop?

Typos
- In Figure 3. being being
- age’s causal effect wages in more than that of nutrition → weighs in more?

It would be desired to have some discussion about the “probability of causation” in causal inference and its relevance to SCI / rule set. These are very related.

===
I updated my score to an accept. I find this paper contains several refreshing and interesting ideas, and the linear and Markovianity assumptions are reasonable for the sake of SCI. Given that the paper provides an interesting idea (which although I feel a little bit preliminary), I wish more researchers read, discuss the paper, and work on making the idea more concrete (e.g., connecting to path-specific effect and counterfactuals).





**Summary Of The Paper:**

This paper investigates how interpretation arises within a DAG structure learned from data. The authors proposed Structural Causal Interpretation (SCI) which provides an explanation (interpretation) given data and a why question based on an assumed structural causal model. Under this definition of “interpretation” or “interpretability”, neural-network induced DAG models are considered interpretable. The authors demonstrated the usefulness of SCI.

**Summary Of The Review:**

While I liked the overall idea of interpretation (which I still think about explanation) of a certain variable having a certain value through causal mechanisms, the paper seems over-promising what interpretation is about with a restricted, simplified causal model.
===after discussion
I liked the overall idea of interpretation (which I still think about explanation) of a certain variable having a certain value through causal mechanisms. Some of the assumptions in the paper seemed too restrict at first, but I later found reasonable for the sake of generating interpretation.

---

> ### Author Response · Authors · 2021-11-12
> **Author's response (Part 1/3)**
>
> Thank you for the review, $\color{purple}{\text{Reviewer 2Fy5}}$!
>
> * > $\color{purple}R:$ "This paper combines causality, interpretation, and machine learning in an intriguing way."
>
>   Unfortunately, this is the only positive aspect that is being mentioned and it is very unspecific/general. Furthermore, the reviewer's summary completely circumvents the interpretability theorem on existing methods (p.7,Thm.3), the connection of mental models and SCMs (p.4, Thm.1, Fig.1), the learning with interpretations for improved sample efficiency (Exp.2, and p.8, Fig.3), and also the human user study  that reveals intriguing properties of human CEM and compares human- versus algorithm-based SCI (Exp.3, and p.9,Fig.4,Tab.2).
>
> * > $\color{purple}R:$ "In page 2, “product distribution over noise variables” and “independent, exogenous variables” imply that the SCM yields a Markovian model, meaning that we observe all the confounders among the measured variables. This is not typically assumed in causal inference literature where causal effect is trivially identified from observational data."
>
>   Respectfully, we disagree both with the statement itself ([1]: Markovianity being non-standard) and with the implied statement ([2]: Markovianity affecting the quality of our work). For [1], the Markovianity is arguably very common in causal discovery/induction literature (to at least give a few examples, consider for instance all the algorithms from Thm.3 like Zheng et al. 2018, Goudet et al. 2018, Yu et al. 2019, Ke et al. 2019 but also more classical works like ICP by Peters et al.; only recently works like Xia et al. 2021 moved beyond Markovianity to integrate reasoning about hidden confounding). For [2], the Markovianity assumption is not of importance for the process of generating interpretations on its own since opposed to causal inference where the causal effect gets confounded when the Markovianity assumption does not hold, this does not apply for interpretations because whether the *content* of the interpretation is correct/real is not relevant but only the *form* of the interpretation i.e., the way it is being generated. To give another example, consider logic. In logic, the form goes over the content. If X is Y and Y is Z implies X is Z, then it does not matter what X, Y, and Z are.
>
> * > $\color{purple}R:$ "Further, the Theorem 1’s definition of structural equation that f_i can be decomposed into an additive equation with no interaction among parents is very strong. This definition is lurking inside theorem as “i.e.” while this assumption is crucial in understanding the precise condition the later theoretical developments rely on and the limitations of the proposed method."
>
>   Said assumption is a linearity assumption and is being pointed out to explicitly in Thm.1 on p.4. Again, such an assumption is neither uncommon in causality-related literature, nor is it of greater significance to the claim/idea of the CMMC Hypothesis (see for instance Fig.1 on p.4). Nonetheless, we gladly consider a re-design of said element.
>
> * > $\color{purple}R:$ "The definition of a single why question seems reasonable but the name doesn’t seem to be specific enough to properly represent what it really is about. Is every why question about the inequality based binary relationship?"
>
>   We appreciate that it is perceived reasonable, and gladly take any more specific, better suited, suggestions for the naming convention. Also, yes, as Def.1 on p.6 explicitly states the considered relations are binary, and this without loss of generality since the relation is being used to assess relative to some population measure how to "classify" any given data point. E.g. considering the lead example, whether Hans's mobility is indeed bad (relative to the population) or not.
>
> ---
> END OF PART 1
>
> PART 2 IN THE FOLLOWING

---

> > ### Author Response · Authors · 2021-11-12
> > **Author's response (Part 2/3)**
> >
> > PART 2:
> >
> > * > $\color{purple}R:$ "Similarly, the name for Proposition 1 "First-Order Logic Rules for Interpretation“ seems a bit broad. Could you please provide any reason why it is a proposition not a definition. (what were claimed and proved?)"
> >
> >   Again, we gladly take any specific suggestions, however, they are First-Order Logic rules and we do in fact give very specific names to the respective rules themselves (excitation, inhibition, and preference). These rule names are arguably natural names which nicely correspond to known concepts from neuroscience (as we expand upon and detail in the appendix A.7 on p.16). Regarding the claim of the proposition, it is to be found at the last sentence of the proposition: For any causal scenario $C_{XY}{:=}(\mathfrak{g}(Y{\rightarrow}X),y,x,\mu_X,\mu_X)$ excitation ($\textbf{R1}$) and inhibition ($\textbf{R2}$) are mutually exclusive. Essentially, it can never occur that a scenario is both excited and inhibited (like it is for any neuron in the brain).
> >
> > * > $\color{purple}R:$ "The authors mentioned that “the three rules in Prop.1 are naturally derived from the principles established in the previous section.“ It would be very helpful if such principles were explicitly mentioned, and how the rules are naturally derived."
> >
> >   The derivation is informally covered in Sections 3.1 and 3.2 p.3-4 and also in the appendix A.7 on p.16. We gladly re-iterate the key principles that led from the informal understanding to the formal interpretation construction in Thm.2 via the rules in Prop.1 both on p.6: [Idea 1] Observe that the why-query contains a relative notion *"why is the property Y of X **bad** ?"* that implicitly compares the individual e.g. in our lead example the attributes of the individual Hans to the questionee's available data set i.e., remaining population - this reveals that perception is linked to the population and thus relative. [Idea 2] observe that the trivial conclusion that by definition there can only exist a causal effect from some variable to another if and only if there is a causal directed path between the variables - this reveals that the interpretation can only act on the causal path of relevance. [Idea 3] consider the difference in causal effect i.e., that some variables exert more influence than others - to give an example, an average car accident has more negative impact on the health of a person than an average workout has positive effect.
> >
> > * > $\color{purple}R:$ "Where do you use the defined metric space? [...] In the middle of page 3. I couldn't fully follow the logic that “Interestingly, this comparability of SCM …” leads to “This insight tells us …”."
> >
> >   We gladly re-iterate on this missed argument. The result on the metric space is being used for two arguments, [1] the comparability of mental models and [2] the relevance of true causality opposed to any. For [1], we reason about the CMMC hypothesis (Causal Mental Model Conversion Hypothesis, Hyp.1 on p.3) which suggests a relation between the human ways of thinking and the formalism of causality's SCM. Achieving the metric space on SCM with Thm.1, allows us now to transitively reason about the comparability of human thinking modes. For [2], given that we have a metric space, we can now compare distances relative to any fixed element in the said space and by this define a notion of "true" causality, and then again, we argue that the true causality is in fact what any ML algorithm seeks for improvement opposed to any causality. I.e., just making a system causal will not help it improve as only the true causal model will (thereby, causality does not matter for performance just for the sake of causality; since arbitrary SCMs can be defined).
> >
> > ---
> > END OF PART 2
> >
> > PART 3 IN THE FOLLOWING

---

> > > ### Author Response · Authors · 2021-11-12
> > > **Author's response (Part 3/3)**
> > >
> > > PART 3:
> > >
> > > * > $\color{purple}R:$ "Why does Theorem 3 exclude any other conditional independence based or score-based causal discovery algorithm which would generate interpretable DAGs (up to its Markov equivalence in some cases) but only NIM-DAGs? Is this because this paper has very restricted assumptions on what an underlying SCM is (linear and Markovian)?"
> > >
> > >   Regarding the first question on Thm.3 and its reach: Thm.3 on p.7 proves that induction/discovery methods such as NOTEARS are capable of generating causal interpretations, and this is relevant since it sheds a completely new light on existing research by implying that all these previous models are indeed interpretable and even in a human-understandable (natural language) manner solely due to their causal nature. Therefore, as correctly pointed out, any causal conditional independence based or score-based method other than the methods we proved Thm.3 for, are likely to work for the same reasons. Since we provide the proof on a case-by-case basis, however, we do not prove the general statement (which we could conjecture at least) but choose arguably popular methods in the current, existing literature that all happen to be neural methodologies.
> > >   Regarding the second question whether the linearity or Markovianity assumptions are relevant: No, neither the linearity nor the Markovianity assumption are of importance for the process of generating interpretations on its own (Thm.2 on p.6) since opposed to causal inference where the causal effect gets confounded when the Markovianity assumption does not hold, and some effects are rendered unidentifiable when the linearity is not satisfied, this does not apply for interpretations because whether the *content* of the interpretation is correct/real is not relevant but only the *form* of the interpretation i.e., the way it is being generated. To give another example, consider logic. In logic, the form goes over the content. If X is Y and Y is Z implies X is Z, then it does not matter what X, Y, and Z are.
> > >
> > > **Conclusive comment from the authors to the review:** We criticize that this review did not consider to point out any strengths of our work, which is an integral part to the reviewing process. Furthermore, the review completely circumvents major parts of the paper, to be specific: the learning with interpretations for improved sample-efficiency (Exp.2, and p.8, Fig.3) and also the human user study that reveals intriguing properties of human CEM and compares human- versus algorithm-based SCI (Exp.3, and p.9,Fig.4,Tab.2).

---

> > > > ### Comment · Reviewer_2Fy5 · 2021-11-18
> > > > **Folllow-up**
> > > >
> > > > Thanks for some clarifications and honest assessment of my review. I focused more on what I felt problematic.
> > > >
> > > > > “the Markovianity assumption is not of importance for the process of generating interpretations on its own …, this does not apply for interpretations because whether the content of the interpretation is correct/real is not relevant but only the form of the interpretation i.e., the way it is being generated”
> > > >
> > > > I am genuinely interested in better understanding the explanation here. Would you kindly provide a simple yet concrete example with two different models e.g., x→y vs. x→y with a hidden confounder X and Y. (e.g., some example like Simpson’s paradox?) Saying that the content of interpretation is irrelevant seems a contradictory statement.
> > > >
> > > >
> > > > >“Again, such an assumption is neither uncommon in causality-related literature, nor is it of greater significance to the claim/idea of the CMMC Hypothesis”
> > > >
> > > > Common or uncommon is not my main point of argument. Here, the question is whether the assumption is crucial (in what aspect). If you can say that it does not play a significant role, why would you put it in the first place.
> > > >
> > > >
> > > > With an impressive title, “Structural Causal Interpretation Theorem” and with unimpressive assumptions about the linearity (additivity) and Markovianity, I don’t feel the paper does not (fully) deliver what’s promised.
> > > > Certainly, I liked the general idea of generating an answer for a why question. But it seems that the derivation of the SCI theorem seems less principled (like CMMC hypothesis and other definitions) and subjective.

---

> > > > > ### Author Response · Authors · 2021-11-18
> > > > > **Author's Response to Follow-up by Reviewer 2Fy5**
> > > > >
> > > > > Thank you for engaging in the discussion, $\color{purple}{\text{Reviewer 2Fy5}}$!
> > > > >
> > > > > * > $\color{purple}R:$ "With an impressive title, “Structural Causal Interpretation Theorem” and with unimpressive assumptions about the linearity (additivity) and Markovianity, I don’t feel the paper does not (fully) deliver what’s promised. Certainly, I liked the general idea of generating an answer for a why question. But it seems that the derivation of the SCI theorem seems less principled (like CMMC hypothesis and other definitions) and subjective."
> > > > >
> > > > >   In the following, regarding the "unimpressive assumptions" perception, we'd like to provide countering argumentation on why our work is invariant to whether said assumptions are impressive or not. [The first bullet covers linearity, the second bullet covers Markovianity.]
> > > > >
> > > > > * > $\color{purple}R:$ "Common or uncommon is not my main point of argument. Here, the question is whether the assumption is crucial (in what aspect). [...]"
> > > > >
> > > > >   The assumption of linearity is crucial in proving the Theorem for the metric space of $n$-sized SCMs (Thm.1 on p.4) and by that allowing us to show the modularity of SCMs and their relation as illustrated in Fig.1 on p.4. For the sake of the CMMC hypothesis, which is untouched by whether we establish a proof for Thm.1 or not, indeed we could've alternatively chosen to conjecture on the existence of such a metric and thereby modularity in the first place.
> > > > >
> > > > > * > $\color{purple}R:$ "I am genuinely interested in better understanding the explanation here. Would you kindly provide a simple yet concrete example with two different models e.g., x→y vs. x→y with a hidden confounder X and Y. (e.g., some example like Simpson’s paradox?) Saying that the content of interpretation is irrelevant seems a contradictory statement."
> > > > >
> > > > >   We gladly re-iterate with a simple, yet concrete example. Nonetheless, please consider Fig.5 in the Appendix (p.20) which covers four different SCMs and their respective SCI for a given why-question. Consider the well-known Simpson's paradox example for the medical setting of Kidney stone treatments from Charig et al., 1986. The setting is given by $T,K,R$ which are Treatment, Kidney Stone Size, and Recovery respectively, and further the graph is given by $T\rightarrow R, K\rightarrow \\{T,R\\}$. It is known that $T=0$ denotes open surgery and $T=1$ denotes Percutaneous nephrolithotomy (being a more involved procedure) and in the overall statistics for recovery of the patient (denoted by $R=1$) we observe $78\\%$ versus $83\\%$  respectively, suggesting that $T=1$ is the better option. Yet, when looking at the confounder $K$ values of patient recovery, we observe $93\\%$ versus $87\\%$ for a small kidney stone $K=0$ and $73\\%$ versus $69\\%$ for a large kidney stone $K=1$ respectively, suggesting that in fact $T=0$ is better instead. This is the paradoxical situation, which is sensible from the causal perspective. If we now ask the single why-question for patient $i$ with say values $T=1,R=0,K=1$ on why $i$ did not recover $r_i<\mu_R$ (where $\mu_R$ is the mean recovery of the data pool), then we obtain an SCI that reads as follows: "Patient $i$ did not recover because of the large kidney stone, although (s)he had Percutaneous nephrolithotomy." Regarding the irrelevance of the Markovianity assumption, consider the non-Markovian alternative to the example where $K$ is a hidden confounder i.e., we only observe $T\rightarrow R$ as the graph. In common settings found in for example Xia et al. 2021, we might at least be aware of the fact that there is hidden confounding present between the two variables and thus have an additional (dashed) bi-directed edge between $T$ and $R$. Let's consider both cases, in the first case, the SCI for the same question as before would read as: "Patient $i$ did not recover although (s)he had Percutaneous nephrolithotomy." We note that simply the reasoning on $K$ is not being delivered, naturally, since $K$ is not in the SCM/CEM that the SCI process observes. For the second case, we'd observe the same reading due to the definition of the SCI construction. Here, however, we note that this case allows for a natural extension of SCI in which the reading could change to possibly, "Patient $i$ did not recover because of *an unknown reason*, although (s)he had Percutaneous nephrolithotomy." Note that this non-Markovian SCI now allows for reasoning with "unknown reasons" since the hidden confounder $K$ will certainly have a causal relation to $R$ (since $K$ is a confounder) but the name of $K$ will not be revealed (since $K$ is hidden). With this example, we thus conclude that Markovianity influences the given information on the underlying SCM and thereby the SCI content but that Markovianity in no way influences/restricts the procedure of generating the SCI in the first place.

---

> > > > > > ### Author Response · Authors · 2021-11-25
> > > > > > **Thank you for the discussion**
> > > > > >
> > > > > > We would like to thank the reviewer for actively engaging in the discussion with us and also increasing the score.

---

### Official Review · Reviewer_bMJP · 2021-11-02

**Correctness:** 2
**Technical Novelty And Significance:** 2
**Empirical Novelty And Significance:** 3
**Recommendation:** 3
**Confidence:** 3

**Main Review:**

The paper generally has a very relevant topic of generating interpretations from DAG models learned from data. It makes interesting
observations, drawing connections between causal inference literature, more traditional DAG learning, and mental models.

Technically, I thought the paper was difficult to follow, in particular in the crucial parts. For example:
- Proposition 1 is very confusing to me. The proposition introduces rules, but these actually seem three properties on a pair R1 and
R2. Where do R1 and R2 come from? Similarly, a ruleset function has a range of {-1,0,1} but I do not see how it relates to the said
properties.
- The main theorem (Theorem 2) I also found confusing. Maybe because of the same reason as in Proposition 1: it mainly seems to define a structural causal interpretation. The actual claim about this definition is quite obvious (i.e., that it 'terminates' if you define it as
a recursive algorithm) if you assume that the pa() function is derived from a DAG.
- Theorem 3: again, I am not sure what is being proved here (and the proof itself does not help much). The only thing that seems to
happen is that it is observed that these algorithms learn SCMs, and therefore a CEM, and therefore can generate an interpretation.
The relevance that some of these models are 'neural' is unclear to me.
I also strongly disagree with the "big picture view" that is given. It is claimed that "any induction performed on the restricted hypothesis space of DAGs is in fact a causal induction". It is well-known that Bayesian network learning algorithm do not necessarily learn causal models. If I understood the paper correctly, then the idea is that you simply learned a "different" causal model compared to the ground truth. In my opinion, this only obfuscates the matter at hand.

In my opinion, the main contribution of this paper is the approach to generate structural causal interpretations from CEMs. This is not completely new however, as there is quite some related work on generating explanations from Bayesian networks. In my opinion, it would
have been more suitable to compare to these type of methods.


**Summary Of The Paper:**

This paper introduces so-called structural causal interpretations. Furthermore, using these interpretations, the authors aim to show that any neural induction method is interpretable. These ideas are illustrated in several experiments.


**Summary Of The Review:**

In my opinion, the contribution is mainly in the definition of structural causal interpretations and how they relate to existing representations. For the moment, I think the contribution should be presented much more clearly in a major revision. I believe that only after this the significance of the results can be assessed much clearly. As a result, I do not recommend acceptance at this point.

---

> ### Author Response · Authors · 2021-11-12
> **Author's response (Part 1/2)**
>
> We hope to clarify with our comments in what we believe to be misconceptions, $\color{purple}{\text{Reviewer bMJP}}$.
>
> * > $\color{purple}R:$ "In my opinion, the contribution is mainly in the definition of structural causal interpretations and how they relate to existing representations. For the moment, I think the contribution should be presented much more clearly in a major revision. I believe that only after this the significance of the results can be assessed much clearly. As a result, I do not recommend acceptance at this point."
>
>   Respectfully, we have to disagree strongly. The pointed out contribution is itself already highly non-trivial. Furthermore, the reviewer's summary completely circumvents the interpretability theorem on existing methods (p.7,Thm.3), the connection of mental models and SCMs (p.4, Thm.1, Fig.1), the learning with interpretations aspect (Exp.2, and p.8, Fig.3), and also the human user study (Exp.3, and p.9,Fig.4,Tab.2).
>
> * > $\color{purple}R:$ "The paper generally has a very relevant topic of generating interpretations from DAG models learned from data. It makes interesting observations, drawing connections between causal inference literature, more traditional DAG learning, and mental models."
>
>   Unfortunately, these very unspecific/general statements are the only positive aspect to be found in the review. Furthermore, we argue that these aspects would be highly non-trivial in the first place.
>
> * > $\color{purple}R:$ "Proposition 1 is very confusing to me. The proposition introduces rules, but these actually seem three properties on a pair R1 and R2. Where do R1 and R2 come from? Similarly, a ruleset function has a range of {-1,0,1} but I do not see how it relates to the said properties."
>
>   This is already defined mathematically, but we gladly re-iterate: $R$ is a binary ordering relation (e.g. $R\in \{<, >\}$) which first appears in the Def.1 (on p.6) and then is substantial in Prop.1 (also on p.6). For the second comment, let's consider rule 1 (**R1**): there are three options available to interact with this rule either the rule does not fire (**R1** returns 0) or the rule does fire (thus returning -1 or 1), and it does so with either the relation tuple $(R_1:=<,R_2:=>)$ or the relation tuple $(R_1:=<,R_2:=>)$ since **R1** requires that $R_1\neq R_2$.
>
> * > $\color{purple}R:$ "The main theorem (Theorem 2) I also found confusing. Maybe because of the same reason as in Proposition 1: it mainly seems to define a structural causal interpretation. The actual claim about this definition is quite obvious (i.e., that it 'terminates' if you define it as a recursive algorithm) if you assume that the pa() function is derived from a DAG."
>
>   We respectfully disagree with both assessments in this comment ([1]: Thm.2 being the main result; [2]: the acceptability of Thm.2 being a theorem). We elaborate. For [1], the interpretations are the core topic of this work but our key results and arguably more important results might be found in grounding principles of mental models and how to connect them to causality's SCMs (p.4, Thm.1, Fig.1), or the relevant logic to the interpretation scheme (p.6, Thm.2), or the interpretability theorem on existing methods (p.7,Thm.3), or the learning with interpretations to improve sample-efficiency (Exp.2, and p.8, Fig.3), or the human user study (Exp.3, and p.9,Fig.4,Tab.2). For [2], we believe we made an adequate choice in posing the result as a theorem since it is important to provide an analysis on the termination requirements in the least (ideally, other properties as well such as correctness or efficiency) and the alternate variant of introducing SCI first as a Definition to then state the termination Theorem seem to introduce unnecessary clutter that decreases readability.
>
> ----
> END OF PART 1
>
> PART 2 IN THE FOLLOWING

---

> > ### Author Response · Authors · 2021-11-12
> > **Author's response (Part 2/2)**
> >
> > PART 2:
> >
> > * > $\color{purple}R:$ "Theorem 3: again, I am not sure what is being proved here (and the proof itself does not help much). The only thing that seems to happen is that it is observed that these algorithms learn SCMs, and therefore a CEM, and therefore can generate an interpretation. The relevance that some of these models are 'neural' is unclear to me. I also strongly disagree with the "big picture view" that is given. It is claimed that "any induction performed on the restricted hypothesis space of DAGs is in fact a causal induction". It is well-known that Bayesian network learning algorithm do not necessarily learn causal models. If I understood the paper correctly, then the idea is that you simply learned a "different" causal model compared to the ground truth. In my opinion, this only obfuscates the matter at hand."
> >
> >   Again, respectfully, we disagree with the statements within this comment ([1]: relevance of Thm.3, [2]: BNs and causality). We elaborate each. For [1], Thm.3 on p.7 proves that induction/discovery methods such as NOTEARS are capable of generating causal interpretations. We believe this result to be very relevant since it sheds a completely new light on existing research by implying that all these previous models are indeed interpretable and even in a human-understandable (natural language) manner solely due to their causal nature. Since we provide the proof on a case-by-case basis, we do not prove the general statement (which we could conjecture at least) but choose arguably popular methods in the existing literature that all happen to be neural methodologies. For [2], it is true that Bayesian Networks are not necessary causal, in fact, Causal Bayesian Networks (CBN) are by definition and the route that our work takes is Algorithm to CEM (which is a sub-property of SCM) to Interpretations which is not a trivial task by any means.
> >
> > **Conclusive comment from the authors to the review:** We criticize that this review did not help us in any way to improve our work, nor did it at all consider to point out any strengths of our work, which is an integral part to the reviewing process. Furthermore, the review completely circumvents major parts of the paper, to be specific: the connection of mental models and SCMs (p.4, Thm.1, Fig.1), the learning with interpretations for improved sample-efficiency (Exp.2, and p.8, Fig.3), and also the human user study that reveals intriguing properties of human CEM and compares human- versus algorithm-based SCI (Exp.3, and p.9,Fig.4,Tab.2).

---

> > > ### Author Response · Authors · 2021-11-24
> > > **Any other questions?**
> > >
> > > We hope we have answered the questions raised in the original review and would be happy to answer any follow up questions that the reviewer might have.

---

> ### Author Response · Authors · 2021-11-28
> **It will be helpful if the reviewer can engage in a discussion!!**
>
> Hello $\color{purple}{\text{Reviewer bMJP}}$,
>
> We had already asked you if there any further questions from your sides or we have alleviated all your concerns on Nov 24th but there was no response. Now there is only 1 day left for discussion, and with the other 2 reviewers being positive about our work, we believe that it will be helpful if you could engage with us in a discussion since we believe that we have answered all your original queries.
>
> Regards,
>
> The Authors

---

### Official Review · Reviewer_T5c7 · 2021-11-05

**Correctness:** 3
**Technical Novelty And Significance:** 3
**Empirical Novelty And Significance:** 2
**Recommendation:** 6
**Confidence:** 3

**Main Review:**

Strengths:
1. The topic studied by the paper is important and novel. It sounds very smart to me to link the two problems (automatic interpretation generation from causal models and improving learning of the causal models using existing interpretations) and study them together.
2. The complete set of theory developed.
3. The findings, especially (1) any neural induction method (for causality learning) is interpretable; (2) SCIs (even if they only approximate interpretations) can improve the learning quality of the neural induction methods.
3. Excellent use of examples for illustration.

Weakness:
1. A main concern I have on the paper is the clarity of its presentation. Although the logic is rigorous, the writing of the paper is not so easy to follow. In many places, the authors have assumed that readers know the meaning of a term used or the purpose of a statement, which leaves readers a lot of guesswork; and some statements are misleading. Some examples:
--Introduction: the (important) term "causal induction method" is used without being introduced.
--Introduction: It is claimed that this paper presents "the first work on causal interpretation" - this is misleading/confusing, since there has been work on providing causal interpretations, e.g. on a decision or prediction by a classifier.
--Theorem 1: what does "n-SCMs" mean? why "d" is defined so?
--Proposition 1: in R1\noteq R2, what are R1 and R2 respectively?
--Table 1: what does ' ' in the diagrams stand for?
--Figure 3: What does each subfigure stand for exactly?
--The 2nd paragraph of Section 4: What are the "algorithmic" interpretations?
Additionally, it may help if in the Introduction, authors could state explicitly the difference between this work and the current focus of the research in "interpretable machine learning", where the goal is to explain a prediction model or the prediction of an instance made by a model.

2. Related work is not complete or lack necessary details. Specifically
-- There has been quite a lot work on incorporating prior knowledge in Bayesian network learning. This should be mentioned as part of related work, since SCIs obtained from human knowledge (or mental process) can be considered as prior knowledge too.
-- Is there any link between SCI and the work presented in the following paper?
Nielsen, U. H., Pellet, J. P., & Elisseeff, A. (2008, July). Explanation trees for causal Bayesian networks. In Proceedings of the Twenty-Fourth Conference on Uncertainty in Artificial Intelligence (pp. 427-434).
-- At the end of Section 2 (page 3), the authors mentioned the work in (Stammer et., 2021). More details should be provided on (Stammer et., 2021) and how it is related to and different from the work in this current paper.

Extra/minor comments:
1. There are some English errors in the paper, e.g.
-- missing articles
-- Page 2, the sentence "An SCM C induces a DAG G with edges ...." is not grammatically correct.
-- Page 4, caption of Figure 1, "... than an average workout has positive)" - please check this sentence. Some grammar problem too.
2. Throughout the paper, the opening double quotation marks `` are not shown correctly.

**Summary Of The Paper:**

This paper proposes a new concept named structural causal interpretation (SCI) and the theory supporting SCI and its applications. Several contributions are made by the paper, including:
(1) The theory and method for automatic generation of human understandable interpretations (i.e. SCIs) from causal models, including deep causal induction models.
(2) The theory and algorithm for using existing SCIs (e.g. obtained from human) to improve the learning of causality by deep causal induction models.
(3) Experiments and the small human study for illustrating the theory and methods.

**Summary Of The Review:**

The research topic is important and new. The theory and method developed looks sound to me, and the findings are interesting and could have potential use.  However, as mentioned above, the readability of the paper should be improved, and more related work or details should be provided.

---

> ### Author Response · Authors · 2021-11-12
> **Author's response (Part 1/2)**
>
> Thank you for your review, $\color{purple}{\text{Reviewer T5c7}}$!
>
> * > $\color{purple}R:$ "The topic studied by the paper is important and novel. It sounds very smart to me to link the two problems (automatic interpretation generation from causal models and improving learning of the causal models using existing interpretations) and study them together."
>
>   We certainly agree and appreciate the shared interest.
>
> * > $\color{purple}R:$ "The complete set of theory developed."
>
>   Indeed, our work establishes itself firstly in the grounding principles of mental models and how to connect them to causality's SCMs (p.4, Thm.1, Fig.1), then develops its relevant theory to lead to the interpretation scheme (p.6, Thm.2), to then be supported by an extensive set of relevant experiments:  interpretability theorem on existing methods (p.7,Thm.3), the connection of mental models and SCMs (p.4, Thm.1, Fig.1), the learning with interpretations aspect (Exp.2, and p.8, Fig.3), and also the human user study (Exp.3, and p.9,Fig.4,Tab.2). Since this is a very strong aspect of our contribution, the reviewer might consider an increase in score.
>
> * > $\color{purple}R:$ "The findings, especially (1) any neural induction method (for causality learning) is interpretable; (2) SCIs (even if they only approximate interpretations) can improve the learning quality of the neural induction methods."
>
>   We certainly agree. The result in (1) sheds a completely new light on existing research, implying that all the previous models are indeed interpretable in a human-understandable manner solely due to their causal nature. Furthermore, (2) offers the foundation for an interactive, human-based way to learning in the future.
>
> * > $\color{purple}R:$ "Excellent use of examples for illustration."
>
>   We appreciate that our goal of clarity of presentation for the newly introduced ideas and concepts in our work seems to be fulfilled.
>
> * > $\color{purple}R:$ "[...] the logic is rigorous [...]"
>
>   We appreciate that also our formalism is set, again, the reviewer might consider an increase of score to reflect this and the other positive assessments.
>
> * > $\color{purple}R:$ "[...] main concern I have on the paper is the clarity of its presentation. [...] Some examples:"
>
>   While we find this statement to contradict the previous, we will cover each example given one-by-one to clarify the misconceptions, since this has been the only relevant concern discussed by the reviewer.
>
>   * > $\color{purple}R:$ "the (important) term "causal induction method" is used without being introduced"
>
>     The paragraph on "Learning Directed Acyclic Graphs." in Sec.2 on p.2 introduces the induction term, which we argue is a standard literature term like the alternative word of "discovery", and prefix words like "neural" or "causal" simply describe what kind of graph is being inducted/discovered.
>
>   * > $\color{purple}R:$ "It is claimed that this paper presents "the first work on causal interpretation" - this is misleading/confusing, since there has been work on providing causal interpretations, e.g. on a decision or prediction by a classifier."
>
>     It seems that the review unintentionally cut off the remainder of the sentence that clarifies this perceived issue! The remainder of the sentence specifies: "—interpreting a (deep) causal induction method by grounding it in its causal semantics, which we call structural causal interpretations (SCI)".
>
>   * > $\color{purple}R:$ "Theorem 1: what does "n-SCMs" mean? why "d" is defined so? "
>
>     This is all defined mathematically directly within Thm.1 on p.4, but we gladly re-iterate: $C_n = \\{ (\mathbf{S},P_{\mathbf{N}}) : |\mathbf{S}|{=}n \\}$, so the $n$ refers to the size of the SCM i.e., the number of nodes and thus structural equations. The goal of Thm.1 has been to establish a metric space on SCM and for this we require to give a constructive proof of such a space (and thereby metric). Defining $d$ based on the absolute norm $|\cdot|$ and the square root of the Jensen-Shannon Divergence allows for a natural metric under additivity (also please refer to the proof in Appendix A.1 on p.13).
>
>   * > $\color{purple}R:$ "Proposition 1: in R1\noteq R2, what are R1 and R2 respectively? "
>
>     Again, this is already defined mathematically, but we re-iterate: $R$ is a binary ordering relation (e.g. $R\in \{<, >\}$) which first appears in the Def.1 (on p.6) and then is substantial in Prop.1 (also on p.6).
>
>   * > $\color{purple}R:$ " Table 1: what does ' ' in the diagrams stand for?"
>
>     It is the "ditto mark" (see https://en.wikipedia.org/wiki/Ditto_mark) that in the context of Tab.1 suggests the same/identical graphs.
>
> -----------------
> END OF PART 1
>
> PART 2 IN THE FOLLOWING

---

> > ### Author Response · Authors · 2021-11-12
> > **Author's response (Part 2/2)**
> >
> > PART 2:
> >
> > * (List from Part 1 continued)
> >   * > $\color{purple}R:$ "Figure 3: What does each subfigure stand for exactly?"
> >
> >     With all due respect, we believed that, the extensive caption clearly defines the contents of the subfigures, but we'll re-iterate: on the left, we see the distribution of errors of the induction method (here, NOTEARS is used) where the green bars are the ones where we additionally learn with existing interpretations (Conj.1 on p.8) and the blue bars just plain induction without interpretations. The middle shows an example, regular induction, while the right shows the exact same setting but with the additional interpretation learning.
> >
> >   * > $\color{purple}R:$ "What are the "algorithmic" interpretations?"
> >
> >     The SCI generated by the (neural) induction algorithms (Thm.3 and Tab.1 on p.7, etc.). To rephrase once more, the CEM (Def.3 on p.7) that is used to generate the SCI stems from the algorithms being used, thus naturally being called algorithmic interpretations. When the CEM come from human, then naturally they are being called human-based SCI (or human interpretations).
> >
> > * > $\color{purple}R:$ "There has been quite a lot work on incorporating prior knowledge in Bayesian network learning. This should be mentioned as part of related work, since SCIs obtained from human knowledge (or mental process) can be considered as prior knowledge too."
> >
> >   We agree that this is a sensible connection, and worthwhile mentioning. We will add this in the paper.
> >
> > * > $\color{purple}R:$ "Is there any link between SCI and the work presented in the following paper? Nielsen, U. H., Pellet, J. P., & Elisseeff, A. (2008, July). Explanation trees for causal Bayesian networks."
> >
> >   In Nielsen et al., 2008 we see a similar origin to the approach of acquiring causal interpretations/explanations i.e., they exploit the semantic nature of BNs in addition to their quantitative knowledge to generate explanations. Apart from the original motivation and place to start, there is little resemblance of our work. Nonetheless, worthwhile mentioning, we'll gladly extend our coverage with this suitable reference.
> >
> > * > $\color{purple}R:$ "the authors mentioned the work in (Stammer et., 2021). More details should be provided on (Stammer et., 2021) and how it is related to and different from the work in this current paper."
> >
> >   The work by Stammer et al., 2021 falls into the XIL category and does not talk about causality in any way. It is very different from our work. Nonetheless, we can add an extended description on the method where space requirements permit.
> >
> > **Conclusive comment from the authors to the review:** We thank the reviewer, again, for the review. The review pointed out several important strengths of the paper, while having only a concern on aspects perceived to be unclear, which we believe have been clear and we nonetheless re-iterated upon one-by-one. Since the score does not reflect this assessment, the reviewer might be inclined to increase the score.

---

> > > ### Author Response · Authors · 2021-11-24
> > > **Any other questions?**
> > >
> > > We hope we have answered the questions raised in the original review and would be happy to answer any follow up questions that the reviewer might have.

---

### Author Response · Authors · 2021-11-18
**Any Further Questions?**

We hope that we have answered all the questions raised by the reviewers. We would be happy to make any further clarifications if required.

We would like to answer any further questions, if any, by the end of the rebuttal period i.e. 29th November.

---

### Comment · Area_Chair_qspc · 2021-11-29
**Some questions and comments to the authors**

Dear Authors,

 In general, I have a positive assessment of the paper, but still have a few issues that followed from the discussion with reviewers and by reading the paper. I would appreciate it if you can share your thoughts on any of them before I make my decision. First, as noted a few times, some critical assumptions are really swept under the rug and should be made explicit. In particular,

1) Additivity is essential and it doesn’t seem easy to relax in any way. While having this assumption is normal in many papers, the particular way of writing seems to be avoiding discussing it, which makes familiar readers uneasy, as mentioned above. It would be expected to make this assumption very clear. Would this be okay with you?

 2) Furthermore, the Markovianity assumption plays a prominent role in the results, even though it may be potentially relaxed. Again, the way the paper is framed and the whole discussion with the reviewers seem to suggest otherwise. Are you committed to make this assumption upfront and explicit in the very beginning, abstract, introduction, and throughout the paper? Please, be specific here.
 I would prefer strongly to see the title changed to “Structural Causal Interpretation Theorem for Additive Markovian Models”. If Markovianity is not key, as claimed, the theory should change accordingly. What is your preference? Markovianity and its violations motivate most of the work in causal inference and cannot be ignored.

 3) The paper contains some good ingredients; in particular, Proposition 1 and Theorem 2, which are its main strengths. I believe this part is interesting and novel enough to be published.

 4) A possible suggestion for improvement would be to relate the points (R1), (R2), (R3) of Proposition 1 to the example of Hans’ health. Specifically, spelling out what the intuitive meaning of these three operators is (in particular, preference is least clear).

 5) It could be made more clear what the “why” questions considered in the paper are, in particular how they differ from previous literature. Also, a discussion about the relation/similarity of the work to actual causes (Halpern & Pearl, 2013) and probabilities of necessity and sufficiency (Greenland & Robins 1999; Tian and Pearl 2000) would help the reader place the work in a more broad context. Are any of these two approaches similar to the proposed work once additivity/Markovianity is assumed? More recently, the (non-parametric) causal explanation formula has been proposed (Zhang and Bareinboim, 2018), is the framework proposed in the paper somewhat related to it (subsumes, implies)?

 6) I am not sure if Theorem 1 and Hypothesis 1 are the major contributions of the paper. The statements made are not supported by mathematical formality — namely “having access to many SCM-encodings of subjective mental models can ultimately lead in their overlap-agreement to (parts of) the objective true causality”. Do the authors think it is possible that the mental models of individuals can overlap but yet not reflect true causality at all? In particular, the scientific community has been many times shown to have incorrect ideas about what the true causality of a model is; having a lot of instances of wrong SCMs that agree could also be misleading and yield an SCM, which is not the true one? I wonder if authors wish to remove, shorten or reconsider this part of the paper to make it factually true. If not removed, explaining a clear connection with Proposition 1 and Theorem 2 would be much desired.

---

> ### Author Response · Authors · 2021-11-29
> **Author's Response to the AC (Part 1/3)**
>
> Thank you for engaging in the discussion, $\color{orange}{\text{Area Chair qspc}}$!
>
> Please find our answers to the points you provided below (chronologically, in the same order):
>
> * > $\color{orange}AC:$ "Additivity is essential and it doesn’t seem easy to relax in any way. While having this assumption is normal in many papers, the particular way of writing seems to be avoiding discussing it, which makes familiar readers uneasy, as mentioned above. It would be expected to make this assumption very clear. Would this be okay with you?"
>
>   **Yes**, this is okay with us.
>
>   To also provide a summary once more on why additivity was not as explicit in the first place: as one can identify from the discussion we shared with $\color{purple}{\text{Reviewer 2Fy5}}$, the assumption of additivity is crucial in proving the Theorem for the metric space of n-sized SCMs (Thm.1 on p.4). It allows us to show the modularity of SCMs and their implied relation as illustrated in Fig.1 on p.4. However, for the sake of the CMMC hypothesis, which remains untouched by whether we establish a proof for Thm.1 or not, indeed we could've alternatively chosen to conjecture on the existence of such a metric and thereby modularity of SCMs in the first place.
>
> * > $\color{orange}AC:$ "Furthermore, the Markovianity assumption plays a prominent role in the results, even though it may be potentially relaxed. Again, the way the paper is framed and the whole discussion with the reviewers seem to suggest otherwise. Are you committed to make this assumption upfront and explicit in the very beginning, abstract, introduction, and throughout the paper? Please, be specific here. I would prefer strongly to see the title changed to “Structural Causal Interpretation Theorem for Additive Markovian Models”. If Markovianity is not key, as claimed, the theory should change accordingly. What is your preference? Markovianity and its violations motivate most of the work in causal inference and cannot be ignored."
>
>   We are **open to both** options:
>
>   * (A) an explicit statement throughout the paper (including the title, abstract etc.)
>   * (B) an adaptation of the theory according to the results from the discussion with $\color{purple}{\text{Reviewer 2Fy5}}$ (i.e., reasoning about "unknown reasons" and the motivational example in a Simpson's paradox setting).
>
>   We **prefer (B)** since we believe that the non-Markovian variant established during discussion is of more value to the scientific community.
>
> * > $\color{orange}AC:$ "The paper contains some good ingredients; in particular, Proposition 1 and Theorem 2, which are its main strengths. I believe this part is interesting and novel enough to be published."
>
>   **Thank you** for the positive assessment and specifically in recognizing the strengths of Prop.1 and Thm.2, which we believe stand at the foundation of  establishing the other presented results that we consider to be important, e.g., interpretability theorem on existing methods (p.7, Thm.3), enforcing consistency with interpretations during learning (Exp.2, and p.8, Fig.3), or the human user study that could hint at a "real-world justification" (Exp.3, and p.9,Fig.4,Tab.2).
>
> * > $\color{orange}AC:$ "A possible suggestion for improvement would be to relate the points (R1), (R2), (R3) of Proposition 1 to the example of Hans’ health. Specifically, spelling out what the intuitive meaning of these three operators is (in particular, preference is least clear).."
>
>   Thank you for the suggestion of improving the reference of Prop.1 to the Hans example. We certainly agree and also believe that this is an important aspect to improve the effectiveness of presentation. In fact, as we pointed out during the discussion with $\color{purple}{\text{Reviewer 2Fy5}}$, we did include an informal treatise in reference to the Hans example. It is being covered in Sections 3.1 and 3.2 p.3-4 (before Prop.1) and also in the appendix A.7 on p.16 (after Prop.1). However, we can improve the clarity of this connection to the intuitive Hans example even further by specifically pointing to the rules and by adding a section right after Prop.1 that stands in direct reference to the example.
>
> ---
> END OF PART 1
>
> PART 2 IN THE FOLLOWING

---

> > ### Author Response · Authors · 2021-11-29
> > **Author's Response to the AC (Part 2/3)**
> >
> > PART 2:
> >
> > * > $\color{orange}AC:$ "It could be made more clear what the “why” questions considered in the paper are, in particular how they differ from previous literature. Also, a discussion about the relation/similarity of the work to actual causes (Halpern & Pearl, 2013) and probabilities of necessity and sufficiency (Greenland & Robins 1999; Tian and Pearl 2000) would help the reader place the work in a more broad context. **[Q1]** Are any of these two approaches similar to the proposed work once additivity/Markovianity is assumed? **[Q2]** More recently, the (non-parametric) causal explanation formula has been proposed (Zhang and Bareinboim, 2018), is the framework proposed in the paper somewhat related to it (subsumes, implies)?."
> >
> >   Thank you for suggesting these specific works as reference points for a broader context. We believe they all belong to the over-arching big picture.
> >
> >   Regarding [**Q1**]: No, neither with additivity/Markovianity, nor without are these approaches similar. In (Halpern & Pearl, 2013), the authors show an improved definition of "actual causes" that shows consistency with various real-world inspired examples (that have troubled the community priorly due to inconsistency with previous definitions). Yet, we believe that the semantics that underly this early work are foundational for our work, and also the application of the definition is inline in nature with the way we present SCI. For PNS (e.g. Greenland & Robins 1999), the difference is arguably more significant since we'd compare a causal quantity $p(y'_{x'},y_x)$ to a causally-rooted algorithm (or simply, a set of causal quantities) in SCI. Yet, we believe PNS capability to express how $y$ could respond to $x$ both ways (assuming binary variables for the moment), might reveal interesting connections to SCI in future research work, since a quantification of a causal effect (see $\mathfrak{g}(\cdot)$ from Prop.1 on p.6) lies at the core of the SCI formalism.
> >
> >   Regarding [**Q2**]: Yes, it is somewhat related (but no, not similar). In (Zhang and Bareinboim, 2018), the Causal Explanation Formula in Thm.1 on p.5 presents a general decomposition of the total variation (which is defined as the conditional distribution difference between $p(y|x_1)-p(y|x_0)$ when observing $X$ change from $x_0$ to $x_1$) into counterfactual spurious, indirect and direct effects - thus being a quantifier of causal effects. Then again, causal effects and their quantification (see $\mathfrak{g}(\cdot)$ from Prop.1 on p.6) lie at the core of the SCI formalism but are distinctively used to dictate the reasoning established via the recursive rule application for generating SCI (Thm.1 on p.6).
> >
> >   We agree and do also believe that pointing out the discussed explicitly is fruitful, which we will do so in the final version. Furthermore, we believe it to be even more important that future research work should consider discussing these types of connections more rigourosly.
> >
> > ---
> > END OF PART 2
> >
> > PART 3 IN THE FOLLOWING

---

> > > ### Author Response · Authors · 2021-11-29
> > > **Author's Response to the AC (Part 3/3)**
> > >
> > > PART 3:
> > >
> > > * > $\color{orange}AC:$ "I am not sure if Theorem 1 and Hypothesis 1 are the major contributions of the paper. The statements made are not supported by mathematical formality — namely “having access to many SCM-encodings of subjective mental models can ultimately lead in their overlap-agreement to (parts of) the objective true causality”. Do the authors think it is possible that the mental models of individuals can overlap but yet not reflect true causality at all? In particular, the scientific community has been many times shown to have incorrect ideas about what the true causality of a model is; having a lot of instances of wrong SCMs that agree could also be misleading and yield an SCM, which is not the true one? I wonder if authors wish to remove, shorten or reconsider this part of the paper to make it factually true. If not removed, explaining a clear connection with Proposition 1 and Theorem 2 would be much desired."
> > >
> > >   Assuming that some true causal model for some phenomenon exists, we agree that the fact that overlap also exists (within different SCM-representations of different human mental models) **does not** guarantee or provide any conclusions on whether the overlap SCM is in fact (even partially) the true causal model. In fact, this is what Section 3.1 (ps.3-4) intends on clarifying. To re-iterate, the key arguments are the CMMC hypothesis (Hyp.1 on p.3), the argument on true causality (p.4), and the comparability that allows for overlap (Thm.1, p.4). In our human user study (ps.9-10, Exp.3: Fig.4,Tab.2, and Appendix ps.23f.), we find empirical evidence of the practicality of overlap i.e., that overlap in humans can lead to an improved causal model that is closer to the ground truth, then again, this is neither a mathematical statement, nor does it allow for making claims on other more complex examples than the "simple" ones considered in study. Generally, for the simple examples one can argue that an implicit assumption exists i.e., the subjects mental models encode a partially correct causal model. We agree, and believe that it is very important to make this discussed line of argumentation very clear. We can achieve this by re-formulating specific segments within Section 3.1 and also as suggested by referencing Prop.1 and Thm.2 in a more direct manner.
> > >
> > > **Conclusive statement:** We thank the $\color{orange}{\text{Area Chair qspc}}$ for helping in improving our work by providing insightful comments and references!

---

### Decision · Program_Chairs · 2022-01-20

**Decision:**

Reject

**Comment:**

The paper contains *fresh* new ideas connecting mental models and SCMs and providing interpretations (explanations) from DAG models learned from data, including those learned by using deep learning. The usefulness of the theory is illustrated with experiments. The paper contributes some theoretical results, but the presentation has serious issues. In general, the reviewers found the paper hard to follow due to a lack of clarity in some notations, definitions, and assumptions.

The paper was discussed in-depth and at length, including the reviewers, the AC, and the senior AC. After all, the gap between the current writing and what is expected from the camera-ready is a bit too large, and we feel it could be a disservice to the authors and community to have the paper accepted in its current form, without passing through another round of reviews. Unfortunately, we do not have any version of "conditional acceptance."

Having said that, we feel the paper has the potential for having a significant impact, and we appreciate the novelty of the proposed approach and the connection among different fields. To avoid issues in the future, we would like to suggest the authors pay attention to the detailed feedback provided by the reviewers, including the discussion and the conversation with the AC, following the exchange on Nov/28. Some examples of points that could make the presentation clearer include 1) clarifying the contributions and providing more examples of the theoretical results, 2) making explicit that the results work for Markovian and additivity models, and 3) perhaps changing the title accordingly.